# RDesign: Hierarchical Data-efficient Representation Learning for Tertiary Structure-based RNA Design

**Cheng Tan**[1,2*], **Yijie Zhang**[3*] **Zhangyang Gao**[1,2*], **Bozhen Hu**[1,2], **Siyuan Li**[1,2],
**Zicheng Liu**[1,2] , **Stan Z. Li**[2†]

[1]Zhejiang University, Hangzhou, China [3]McGill University, Montréal, Québec, Canada
[2]AI Lab, Research Center for Industries of the Future, Westlake University, Hangzhou, China
{tancheng,gaozhangyang}@westlake.edu.cn; yj.zhang@mail.mcgill.ca

## Abstract

While artificial intelligence has made remarkable strides in revealing the relationship between biological macromolecules' primary sequence and tertiary structure, designing RNA sequences based on specified tertiary structures remains challenging. Though existing approaches in protein design have thoroughly explored structure-to-sequence dependencies in proteins, RNA design still confronts difficulties due to structural complexity and data scarcity. Moreover, direct transplantation of protein design methodologies into RNA design fails to achieve satisfactory outcomes although sharing similar structural components. In this study, we aim to systematically construct a data-driven RNA design pipeline. We crafted a large, well-curated benchmark dataset and designed a comprehensive structural modeling approach to represent the complex RNA tertiary structure. More importantly, we proposed a hierarchical data-efficient representation learning framework that learns structural representations through contrastive learning at both cluster-level and sample-level to fully leverage the limited data. By constraining data representations within a limited hyperspherical space, the intrinsic relationships between data points could be explicitly imposed. Moreover, we incorporated extracted secondary structures with base pairs as prior knowledge to facilitate the RNA design process. Extensive experiments demonstrate the effectiveness of our proposed method, providing a reliable baseline for future RNA design tasks. The source code and benchmark dataset are available at github.com/A4Bio/RDesign.

## 1 Introduction

Ribonucleic acid (RNA) is a fundamental polymer composed of ribonucleotides, serving as a vital biological macromolecule that regulates a plethora of cellular functions (Kaushik et al., 2018; Guo et al., 2010; Sloma & Mathews, 2016; Warner et al., 2018). Non-coding RNA strands exhibit intricate three-dimensional structures, which are essential for their biological activities (Feingold & Pachter, 2004; Gstir et al., 2014). The complex geometries of RNA molecules empower them to execute irreplaceable functions in crucial cellular processes (Crick, 1970), encompassing but not limited to mRNA translation (Roth & Breaker, 2009), RNA splicing (Runge et al., 2018; Wanrooij et al., 2010; Kortmann & Narberhaus, 2012), and gene regulation (Meyer et al., 2016).

Specifically, the primary structure of RNA refers to its linear sequence of ribonucleotides (Hofacker et al., 1994; Rother et al., 2011; Kagaya et al., 2023). The primary structure then folds into a secondary structure with canonical base pairs, forming stems and loops (Nicholas & Zuker, 2008; Yang et al., 2017; Liu et al., 2022). Tertiary interactions between secondary structural elements subsequently give rise to the three-dimensional structure (Qin et al., 2022; Wang & Dokholyan, 2022; Yesselman & Das, 2015; Das et al., 2010). Figure 1 illustrates an example of the hierarchical folding of RNA primary, secondary, and tertiary structures. Gaining a comprehensive understanding

---

*Equal contribution.
†Corresponding author.

of RNA structure is fundamental to figuring out biological mysteries and holds tremendous promise for biomedical applications. However, solving RNA structures through experimental techniques remains challenging due to their structural complexity and transient nature. Computational modeling of RNA structure and dynamics has thus become particularly valuable and urgent.

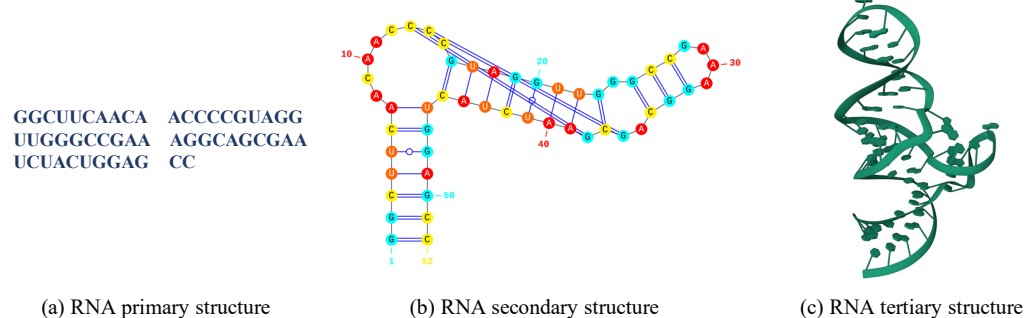

| | | |
|---|---|---|
| (a) RNA primary structure | (b) RNA secondary structure | (c) RNA tertiary structure |

Figure 1: The schematic diagrams of RNA primary, secondary and tertiary structures.

Recent years have witnessed the emergence and rapid advancement of data-driven computational modeling of RNA (Angermueller et al., 2016; Xiong et al., 2021; Singh et al., 2021; Cao et al., 2024). In particular, algorithms for RNA secondary structure prediction have been extensively developed, yielding impressive results through leveraging large datasets of known secondary structures (Singh et al., 2019; Chen et al., 2019b; Fu et al., 2022; Tan et al., 2022). However, knowledge of RNA tertiary structures, which is crucial for thoroughly understanding RNA functional mechanisms and discovering RNA-targeted therapies (Warner et al., 2018; Churkin et al., 2018), remains limited (Townshend et al., 2021). The success of protein structure prediction (Jumper et al., 2021; Baek et al., 2021) has motivated researchers to tackle the even more challenging problem of RNA tertiary structure prediction, leading to the development of RNA tertiary structure folding algorithms such as DeepFoldRNA (Pearce et al., 2022), RoseTTAFoldNA (Baek et al., 2022), and RhoFold (Chen et al., 2022; Shen et al., 2022). While predicting RNA tertiary structures from primary sequences can leverage abundant sequence data (Chen et al., 2022), its inverse problem, designing RNA sequences that reliably fold into a specified tertiary structure, remains largely underexplored.

The key reasons that RNA tertiary structure modeling lags far behind protein tertiary structure modeling stem from two main aspects: (1) *RNA demonstrates greater structural intricacy and flexibility than proteins, posing formidable hurdles for structure prediction and design* (Townshend et al., 2021; Bernstein et al., 2012; Berman et al., 2000). The less constrained structure space of RNA leads to intractable challenges in modeling the RNA tertiary structure. (2) *High-resolution RNA tertiary structures are scarce compared to proteins due to their conformational dynamics and instability*(Rother et al., 2011). The quantity of available RNA structures constitutes less than 1% of that for proteins (Adamczyk et al., 2022b; Kalvari et al., 2021). To deal with the above problem, we propose a thorough pipeline aiming at data-driven tertiary structure-based RNA design tasks. In detail, we first compile a large-scale RNA tertiary structure dataset based on extant high-quality structure data from Protein Data Bank (PDB) (Berman et al., 2000) and RNAsolo (Adamczyk et al., 2022b). Then, regarding the deficiency in RNA tertiary structure modeling and the unsatisfying transferable capability of conventional protein structure modeling techniques to the RNA field, we propose a comprehensive RNA tertiary structure modeling. To optimize the use of the limited data, we introduce a hierarchical and data-efficient representation learning framework that applies contrastive learning at both the cluster and sample levels. We can explicitly impose intrinsic relationships between the data by constraining the data representations within a limited hyperspherical space. Moreover, we provide a strategy that utilizes extracted secondary structure as prior information to guide the RNA design inspired by the correlation between RNA secondary and primary structures.

The main contributions of this work are summarized as follows:

- We propose a formal formulation of the tertiary structure-based RNA design problem. To establish a fair benchmark for tertiary structure-based RNA design, we compile a large dataset of RNA tertiary structures and provide a fundamental data split based on both structural similarity and sequence length distribution.

- We propose a comprehensive structural modeling approach for the complex RNA tertiary structure and design an RNA design framework called *RDesign*, which is composed of a hierarchical representation learning scheme and a secondary structure imposing strategy.

- Through extensive experiments across standard RNA design benchmarks and generalization ability assessments, we demonstrate the efficacy of our proposed method. This provides a reliable pipeline for future research in this important and promising field.

## 2 RELATED WORK

### 2.1 BIOMOLECULAR ENGINEERING

In recent decades, the rapid advancements in biophysics, biochemistry, and chemical engineering have enabled a plethora of novel applications (Nagamune, 2017), including engineering enzymes for industrial biocatalysis (Pugh et al., 2018), tailoring antibodies for precision cancer therapies (Jeschek et al., 2016), developing trackable fluorescent proteins for biological imaging (Rosenbaum, 2017), and optimizing polymerases for forensic DNA analysis (Martell et al., 2016). RNA design is of particular interest among them due to the diverse functions that RNA can fulfill, ranging from translation and gene expression to catalysis (Ellefson et al., 2016). This multifunctionality is ascribed to the structural diversity of RNA (Andronescu et al., 2004). In this work, we focus on tertiary structure-based RNA design to uncover the relationships between RNA structure and sequence.

### 2.2 PROTEIN DESIGN

RNA and protein are essential components of cells. Despite having different chemical constituents, their higher-order structures can be described similarly (Rother et al., 2011). Early works on computational protein design (Wang et al., 2018; Chen et al., 2019a) utilize multi-layer perceptron (MLP), and convolutional neural network(CNN) to predict residue types from protein structure. 3D CNNs have enabled tertiary structure-based design such as ProDCoNN (Zhang et al., 2020) and DenseCPD (Qi & Zhang, 2020). GraphTrans (Ingraham et al., 2019) combines attention (Vaswani et al., 2017) and auto-regressive decoding to generate protein sequences from graphs, inspiring a series of recent advancing approaches (Jing et al., 2020; Dauparas et al., 2022; Hsu et al., 2022; Tan et al., 2023; Gao et al., 2022b; 2023; 2024; Tan et al., 2024). While insights from protein research have illuminated RNA biology, RNA studies have trailed due to a scarcity of available data and complex structure modeling (Gan et al., 2003).

### 2.3 RNA DESIGN

The computational design of RNA sequences aims to generate nucleic acid strands that will fold into a targeted secondary or tertiary structure. Secondary structure-based RNA design was first introduced by Vienna (Hofacker et al., 1994). Early works solved the RNA design problem through stochastic optimization and energy minimization with thermodynamic parameters, such as RNAfold (Lorenz et al., 2011), Mfold (Zuker, 2003), UNAFold (Nicholas & Zuker, 2008), and RNAStructure (Mathews, 2014). Probabilistic models and posterior decoding were employed to solve this problem (Sato et al., 2009). Other works that operate on a single sequence and try to find a solution by changing a few nucleotides include RNAInverse (Hofacker et al., 1994), RNA-SSD (Andronescu et al., 2004), INFO-RNA (Busch & Backofen, 2006), and NUPACK (Zadeh et al., 2011). There are global searching methods, including antaRNA (Kleinkauf et al., 2015), aRNAque (Merleau & Smerlak, 2022), eM2dRNAs (Rubio-Largo et al., 2023) and MCTS-RNA (Yang et al., 2017). Reinforcement learning-based methods have also been developed (Runge et al., 2018).

Although numerous approaches have been studied for engineering RNA secondary structures, RNA design based on the tertiary structure is still challenging due to the lack of high-resolution structural data (Yesselman & Das, 2015; Das et al., 2010). Although structure prediction algorithms (Liu et al., 2022; Qin et al., 2022; Wang & Dokholyan, 2022) can utilize abundant RNA primary sequence information, the progress of RNA design has been hindered by the scarcity of determined RNA 3D structures. To alleviate the difficulty, we explore the uncharted areas of tertiary structure-centric RNA design systematically and propose a complete pipeline to address this challenge.

## 3 METHODS

### 3.1 PRELIMINARIES

For an RNA sequence in its primary structure, we assume it comprises $N$ nucleotide bases selected from the set of nucleotides: A (*Adenine*), U (*Uracil*), C (*Cytosine*), and G (*Guanine*). Therefore, the sequence can be represented as:

$$\text{Nucleotides} := \{\text{A, U, C, G}\},$$
$$\mathcal{S}^N = \{s_i \in \text{Nucleotides} \mid i \in [1, N] \cap \mathbb{Z}\}, \tag{1}$$

The formation of the tertiary structure requires the folding of this sequence in three-dimensional space, which can be denoted as:

$$\text{Atoms} := \{\text{P}, \text{O}5', \text{C}5', \text{C}4', \text{C}3', \text{O}3'\},$$
$$\mathcal{X}^N = \{\boldsymbol{x}_i^\omega \in \mathbb{R}^3 \mid i \in [1, N] \cap \mathbb{Z}, \omega \in \text{Atoms}\}, \tag{2}$$

where the Atoms set denotes the six atoms that comprise the RNA backbone.

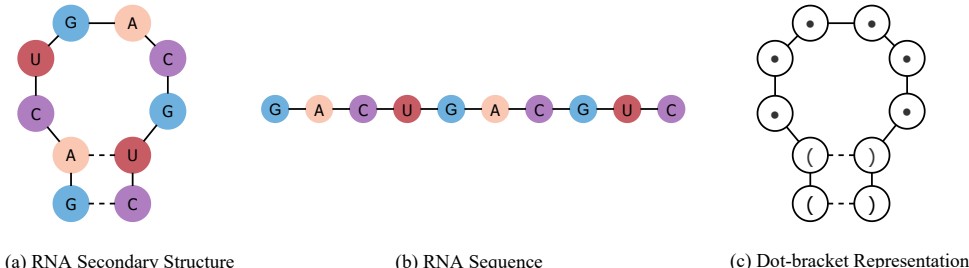

(a) RNA Secondary Structure        (b) RNA Sequence        (c) Dot-bracket Representation

Figure 2: Brief view of RNA sequence and secondary structure.

We incorporate secondary structure information using dot-bracket notation. Unpaired nucleotides are represented by dots, and paired nucleotides are represented by brackets, as shown in Figure 2.

$$\mathcal{A}^N = \{a_i \in \{\ .\ ,\ (\ ,\ )\ \} \mid i \in [1, N] \cap \mathbb{Z}\}, \tag{3}$$

where $a_i$ is a dot if the nucleotide is unpaired, or a matching bracket otherwise. Finally, the tertiary structure-based RNA design problem could be formulated as:

$$\mathcal{F}_\Theta : \mathcal{X}^N \mapsto \mathcal{S}^N,$$
$$\text{such that } \mathcal{A}^N = g(\mathcal{X}^N) \text{ satisfies the pairing rules,} \tag{4}$$

where $\mathcal{F}_\Theta$ is a learnable mapping with parameters $\Theta$, and $g(\cdot)$ is a function that extracts the secondary structure denoted by dot-bracket notation from the tertiary structure. Namely, $\mathcal{F}_\Theta$ denotes the mapping from $\mathcal{X}^N$ to $\mathcal{S}^N$, which means from the tertiary structure to the primary structure. It illustrates that while we map the tertiary structure $\mathcal{X}$ of RNA to the primary sequence $\mathcal{S}$, we ensure that the predicted sequence aligns with the pairing rules associated with the secondary structure $\mathcal{A}$.

### 3.2 COMPREHENSIVE RNA TERTIARY STRUCTURE MODELING

We construct a local coordinate system $\boldsymbol{Q}_i$ for the $i$-th nucleotide in the RNA tertiary structure. The detailed procedure for defining the local coordinate system is in Appendix D. While studies on protein design have achieved considerable success using only the C$\alpha$ atoms to model backbone geometry, this approach does not readily translate to RNA. RNA exhibits a diversity of backbone conformations and base-pairing geometries that cannot be sufficiently captured by such modeling. The complexity and plasticity of RNA structure necessitate a comprehensive treatment.

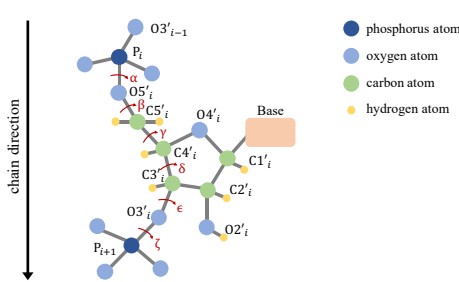

Figure 3: An example of RNA structure.

To adequately capture the complex structural information inherent in the three-dimensional folding of RNA molecules, we propose a general approach for modeling RNA tertiary structure. We represent RNA tertiary structure as an attributed graph $\mathcal{G} = (V, E)$ comprising node attributes $V$ and edge attributes $E$. The graph is constructed by identifying the $K$ nearest neighbors in 3D space for each node; each node $i$ has a set of $K$ neighbors denoted $\mathcal{N}(i, K)$. Specifically, $V \in \mathbb{R}^{N \times f_n}$ contains $f_n$-dimensional node attributes for $N$ nodes, and $E \in \mathbb{R}^{N \times K \times f_m}$ contains $f_m$-dimensional edge attributes for each node's $K$ neighbors. By default, we set $K = 30$.

We outline the attributes used in our modeling approach along with their corresponding illustrations in Table 1, which includes two levels of attributes: (i) intra-nucleotide level attributes describing the local geometry of each nucleotide as the node attribute $V$, and (ii) inter-nucleotide level attributes describing the relative geometry between nucleotides as the edge attribute $E$.

**Intra-nucleotide level** (1) The dihedral angles, shown as red arrows in Figure 3, are calculated. We represent the dihedral angles of the RNA backbone using $\sin$ and $\cos$ functions. (2) The spatial distances between *the other intra-nucleotide atoms* and the atom $P_i$ are encoded into radial basis functions (RBFs). (3) The directions of the other intra-nucleotide atoms relative to the atom $P_i$ are calculated with respect to the local coordinate system $\boldsymbol{Q}_i$.

**Inter-nucleotide level** (1) An orientation encoding $\mathbf{q}(\cdot)$ is calculated from the quaternion representation of the spatial rotation matrix $\mathbf{Q}_i^T \mathbf{Q}_j$. (2) The spatial distances between inter-nucleotide atoms from *neighboring nucleotides* and the atom $P_i$ are encoded into radial basis functions (RBFs). (3) The directions of the other inter-nucleotide atoms relative to the atom $P_i$ are calculated.

Table 1: The feature construction of RNA tertiary structure modeling.

| Level | Feature | Illustration |
|---|---|---|
| | Dihedral Angle | $\big\{ \sin, \cos \big\} \times \big\{ \alpha_i, \beta_i, \gamma_i, \delta_i, \epsilon_i, \zeta_i \big\}$ |
| Intra-nucleotide | Distance | $\Big\{ \mathrm{RBF}(\|\omega_i - P_i\|) \,\Big|\, \omega \in \{O5', C5', C4', C3', O3'\} \Big\}$ |
| | Direction | $\Big\{ \mathbf{Q}_i^T \frac{\omega_i - P_i}{\|\omega_i - P_i\|} \,\Big|\, \omega \in \{O5', C5', C4', C3', O3'\} \Big\}$ |
| | Orientation | $\mathbf{q}(\mathbf{Q}_i^T \mathbf{Q}_j)$ |
| Inter-nucleotide | Distance | $\Big\{ \mathrm{RBF}(\|\omega_j - P_i\|) \,\Big|\, j \in \mathcal{N}(i, K), \omega \in \{O5', C5', C4', C3', O3'\} \Big\}$ |
| | Direction | $\Big\{ \mathbf{Q}_i^T \frac{\omega_j - P_i}{\|\omega_j - P_i\|} \,\Big|\, j \in \mathcal{N}(i, K), \omega \in \{O5', C5', C4', C3', O3'\} \Big\}$ |

## 3.3 Hierarchical Data-efficient Representation Learning

Now that RNA tertiary structure has been adequately modeled, the remaining challenge is how to learn from scarce data in a data-efficient manner. The key motivation is explicitly imposing the inherent data relationships based on prior knowledge. We first use $L$ layers of message-passing neural networks (MPNNs) to learn the node representation. Specifically, the $l$-th hidden layer of the $i$-th nucleotide is defined as follows:

$$\boldsymbol{h}_{V_i}^{(l)} = \mathrm{MPNN}([\boldsymbol{h}_{E_{ij}}, \boldsymbol{h}_{V_i}^{(l-1)}, \sum_{j \in \mathcal{N}(i,K)} \boldsymbol{h}_{V_j}^{(l-1)}]), \tag{5}$$

where $\boldsymbol{h}_V^{(0)}$, $\boldsymbol{h}_E$ are the embeddings of the intra-nucleotide and inter-nucleotide level features from the tertiary structure modeling, respectively. When generating the RNA sequence, a fully connected layer $f$ maps the node representation $\mathbf{h}_{V_i}^{(L)}$ to the RNA sequence space: $f(\mathbf{h}_{V_i}^{(L)})$.

To enable data-efficient representation learning, we obtain the graph-level representation through the average pooling of the node representations $\boldsymbol{h}_G = \frac{1}{N} \sum_{i=1}^{N} \boldsymbol{h}_{V_i}^{(L)}$ and the corresponding projection $g(\boldsymbol{h}_G)$ by the projection $g : \mathbb{R} \to \mathbb{S}$ that projects the Euclidean space into the hyperspherical space.

We propose a hierarchical representation learning framework comprising cluster-level and confidence-aware sample-level representation learning, as shown in Figure 4. The cluster-level rep-

resentation learning utilizes topological similarity between RNA structures. We obtain RNA structure clusters based on TM-score, which indicates structural similarity(Zhang & Skolnick, 2005). We define positive pairs as RNA data with similar topological structures that attract each other in the embedding space, while negative pairs with dissimilar topological structures repel each other. The cluster-level representation learning is defined as follows:

$$\mathcal{L}_{cluster} = -\sum_{p\in\mathcal{D}} \log \frac{\exp(g_p \cdot g_q/\tau)}{\sum_{g_k\in\{g_q\}\cup\mathcal{K}_c} \exp(g_p \cdot g_k/\tau)}, \tag{6}$$

where $(g_p, g_q)$ is a positive pair that comes from the same structural cluster, $\mathcal{K}_c$ is a set of negative samples for $g_p$ identified by the cluster they belong to, and $\mathcal{D}$ is the data set. We denote $g_p$ as the graph representation projection of the $p$-th RNA sample for notational convenience.

The confidence-aware sample-level representation learning is designed to capture the microscopic intrinsic properties of RNA structures. The positive pairs are defined as a given RNA structure sample and its random perturbed structures. The perturbed structures are obtained by adding Gaussian noise to the experimentally determined coordinates. To prevent excessive deviation, we filter out the perturbed structures with low structural similarity (TM-score $\leq 0.8$) and high structural deviation (RMSD $\geq 1.0$). The RMSD also evaluates the confidence level of the perturbed data. Formally, the sample-level representation learning can be formulated as:

$$\mathcal{L}_{sample} = -\sum_{p\in\mathcal{D}} \gamma_{p,p'} \log \frac{\exp(g_p \cdot g_p'/\tau)}{\sum_{g_k\in\{g_p'\}\cup\mathcal{K}_s} \exp(g_p \cdot g_k/\tau)}, \tag{7}$$

where $p'$ is the perturbed structure of the $p$-th RNA structure, and $\mathcal{K}_s$ is simply defined as other samples apart from $g_p$. The confidence score $\gamma_{p,p'}$ is defined as $\exp^{-\text{RMSD}(p,p')}$ so that when $\text{RMSD}(p, p') \to 0$, the confidence approaches 1.

The cluster-level representation provides a coarse-grained embedding, capturing the global topological similarity between RNA structures. The confidence-aware sample-level representation provides intrinsic knowledge that is robust to minor experimental deviations. As shown in Figure 4, by constraining the limited data into the restricted hyperspherical space with imposed prior knowledge, the intrinsic relationships between data are explicitly modeled.

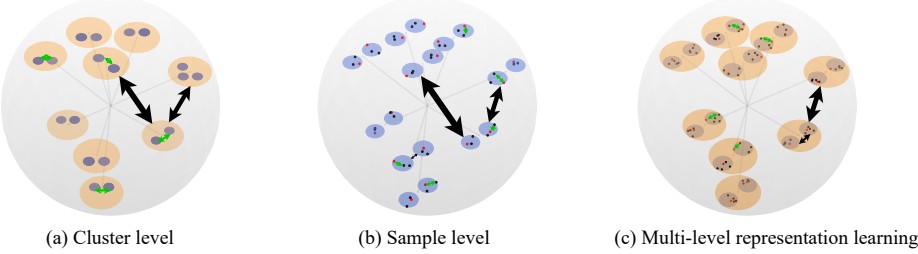

(a) Cluster level       (b) Sample level       (c) Multi-level representation learning

Figure 4: The different levels of hierarchical representation learning. Green arrows denote positive pairs tend to attract each other, and the black arrow denotes negative pairs tend to repel each other.

### 3.4 SECONDARY STRUCTURE IMPOSING STRATEGY

With the given tertiary structure, we can derive the corresponding secondary structure using the notation shown in Figure 2. The secondary structure is represented using parentheses to indicate paired nucleotides, with unpaired sites represented by one of the four RNA nucleotides (A, U, C, G). Paired sites are denoted by two nucleotides placed simultaneously in one of the following pairs: {CG, GC, AU, UA, UG, GU}. When a pair of positions $(i, j)$ in the predicted primary sequence and its corresponding secondary structure are given, we can calculate the confidence score for each position $i$ based on the predicted letter at that position and the known secondary structure constraint. We then choose the position with the higher confidence score as the "reference" (say position $i$). We correct the predicted letter at position $j$ so that the letters at $i$ and $j$ form the allowed pairs.

Specifically, if position $i$ is selected as the reference, we maintain the predicted letter at $i$ unchanged and modify the predicted letter at $j$ to satisfy the base pairing constraint. We then update the predicted primary sequence. By leveraging the information from the known secondary structure, we

can rectify and refine the initially predicted primary sequence. The refinement helps enhance the accuracy of RNA 3D structure prediction. In the training phase, we compel the model to sharpen the confidence of the nucleotides in the paired positions. The supervised loss is defined as:

$$\mathcal{L}_{sup} = \sum_{(i,j)\in\text{Pairs}} \left[\ell_{CE}(s_i, f(\boldsymbol{h}_{V_i}^{(L)})/\tau') + \ell_{CE}(s_j, f(\boldsymbol{h}_{V_j}^{(L)})/\tau')\right] + \sum_{k\notin\text{Pairs}} \ell_{CE}(s_k, f(\boldsymbol{h}_{V_k}^{(L)})), \quad (8)$$

where Pairs encompasses all the paired position indices given by the secondary structure, $\tau'$ is the temperature that is set as $0.5$ by default to sharpen the confidence of paired nucleotides, $s_i$ are true sequence labels.

The training objective is the linear combination of representation learning loss and supervised loss:

$$\mathcal{L} = \mathcal{L}_{sup} + \lambda(\mathcal{L}_{cluster} + \mathcal{L}_{sample}), \quad (9)$$

where we set the weight parameter $\lambda$ as $0.5$ by default.

## 4 EXPERIMENTS

We evaluate RDesign on the tertiary structure-based RNA design task by comparing it with four categories of baseline models: (i) sequence-based models (SeqRNN and SeqLSTM) that do not utilize any structural information and could be viewed as the performance reference for RNA design; (ii) A tertiary structure-based model (StructMLP) that exploits structural features while ignoring the graph topological structure; (iii) Tertiary structure-based models (StructGNN and GraphTrans) (Ingraham et al., 2019) and PiFold (Gao et al., 2022b) that incorporate the graph topological structure; (iv) secondary structure-based RNA sequence design models (MCTS-RNA (Yang et al., 2017), LEARNA (Runge et al., 2018), eM2dRNAs (Rubio-Largo et al., 2023), aRNAque (Merleau & Smerlak, 2022)). The detailed experimental settings and dataset descriptions are shown in Appendix A and Appendix B-C, respectively.

### 4.1 STANDARD TERTIARY STRUCTURE-BASED RNA DESIGN

Using our carefully curated benchmark dataset, we trained the model using the training set. Then, we evaluated the performance of the model on the testing set by selecting the model with the lowest loss on the validation set. Given that RNA sequences of varying lengths may impact prediction results, we stratified the testing set into three groups based on RNA length: (i) *Short*, RNA samples less than or equal to 50 nucleotides; (ii) *Medium* - RNA samples greater than 50 nucleotides but less than or equal to 100 nucleotides; (iii) *Long* - RNA samples greater than 100 nucleotides. To gain a thorough understanding of the relationship between RNA length and model accuracy, we reported both the recovery and Macro-F1 metrics for Short, Medium, and Long testing samples separately, in addition to overall testing set performance.

Table 2: The recovery on the benchmark dataset. The best results are highlighted in bold.

| Method | Recovery (%) ↑ | | | |
| --- | --- | --- | --- | --- |
| | Short | Medium | Long | All |
| SeqRNN (h=128) | 26.52±1.07 | 24.86±0.82 | 27.31±0.41 | 26.23±0.87 |
| SeqRNN (h=256) | 27.61±1.85 | 27.16±0.63 | 28.71±0.14 | 28.24±0.46 |
| SeqLSTM (h=128) | 23.48±1.07 | 26.32±0.05 | 26.78±1.12 | 24.70±0.64 |
| SeqLSTM (h=256) | 25.00±0.00 | 26.89±0.35 | 28.55±0.13 | 26.93±0.93 |
| StructMLP | 25.72±0.51 | 25.03±1.39 | 25.38±1.89 | 25.35±0.25 |
| StructGNN | 27.55±0.94 | 28.78±0.87 | 28.23±1.95 | 28.23±0.71 |
| GraphTrans | 26.15±0.93 | 23.78±1.11 | 23.80±1.69 | 24.73±0.93 |
| PiFold | 24.81±2.01 | 25.90±1.56 | 23.55±4.13 | 24.48±1.13 |
| RDesign | **37.22**±1.14 | **44.89**±1.67 | **43.06**±0.08 | **41.53**±0.38 |

As presented in Table 2, the baseline models achieved suboptimal recovery scores, with performance ranging from 24-28%. Unexpectedly, tertiary structure-based models like StructMLP, StructGNN, and GraphTrans attained comparable results to sequence-based models. This indicates that directly applying protein design techniques to RNA is misguided and fails to capture the intricacies of RNA structures. Moreover, StructMLP and GraphTrans achieved higher recovery scores for short RNA

sequences but struggled on longer, namely more complex RNA structures. Their struggles on the long dataset stem from the inability to learn from more intricate RNA structures and their low generalization capability. In contrast, our RDesign model outperforms all baseline methods on the recovery metric, achieving substantial gains. RDesign's strong performance, particularly on medium and long sets, indicates that it can learn intrinsic RNA structural properties.

Table 3: The Macro-F1 on the benchmark dataset. The score is multiplied by 100 for aesthetics.

| Method | Macro F1 ($\times$100) $\uparrow$ | | | |
| --- | --- | --- | --- | --- |
| | Short | Medium | Long | All |
| SeqRNN (h=128) | 17.22$\pm$1.69 | 17.20$\pm$1.91 | 8.44$\pm$2.70 | 17.74$\pm$1.59 |
| SeqRNN (h=256) | 12.54$\pm$2.94 | 13.64$\pm$5.24 | 8.85$\pm$2.41 | 13.64$\pm$2.69 |
| SeqLSTM (h=128) | 9.89$\pm$0.57 | 10.44$\pm$1.42 | 10.71$\pm$2.53 | 10.28$\pm$0.61 |
| SeqLSTM (h=256) | 9.26$\pm$1.16 | 9.48$\pm$0.74 | 7.14$\pm$0.00 | 10.93$\pm$0.15 |
| StructMLP | 17.46$\pm$2.39 | 18.57$\pm$3.45 | 17.53$\pm$8.43 | 18.88$\pm$2.50 |
| StructGNN | 24.01$\pm$3.62 | 22.15$\pm$4.67 | 26.05$\pm$6.43 | 24.87$\pm$1.65 |
| GraphTrans | 16.34$\pm$2.67 | 16.39$\pm$4.74 | 18.67$\pm$7.16 | 17.18$\pm$3.81 |
| PiFold | 17.48$\pm$2.24 | 18.10$\pm$6.76 | 14.06$\pm$3.53 | 17.45$\pm$1.33 |
| RDesign | **38.25**$\pm$3.06 | **40.41**$\pm$1.27 | **41.48**$\pm$0.91 | **40.89**$\pm$0.49 |

It could be seen from Table 3 that there exist large gaps between the recovery metrics and Macro-F1 scores of most baseline models, which suggests those models tend to predict the high-frequency nucleotide letters instead of reflecting the actual tertiary structure. Among them, only StructGNN achieved consistent results in its Macro-F1 score and recovery metric but with unsatisfying performance. Our proposed RDesign consistently outperformed all other models on these metrics, demonstrating its effectiveness.

## 4.2 EVALUATE THE GENERALIZATION ON RFAM AND RNA-PUZZLES

To assess the generalization capability of our model, we evaluated our model and the baseline methods on the Rfam (Kalvari et al., 2021) and RNA-Puzzles (Miao et al., 2020) datasets using the model pre-trained on our benchmark training set. We presented the results in Table 4. The performance remained consistent with that of our benchmark dataset. Specifically, StructGNN, which effectively learned certain tertiary structure information, achieved a relatively small gap between the recovery metric and Macro F1 score. In contrast, the other baselines that learned little structural information performed sub-optimally. Our proposed RDesign model demonstrated superior generalization on both datasets and outperformed all the baselines.

It is notable that the results reported here were generated by directly assessing pretrained models on the entire training dataset, mirroring real-world scenarios. Furthermore, in Appendix H, we have included the results of the pretrained models assessed on a training set with similar data removed.

Table 4: The overall recovery and Macro-F1 scores on the Rfam and RNA-Puzzles datasets.

| Method | Recovery (%) $\uparrow$ | | Macro F1 ($\times$100) $\uparrow$ | |
| --- | --- | --- | --- | --- |
| | Rfam | RNA-Puzzles | Rfam | RNA-Puzzles |
| SeqRNN (h=128) | 27.99$\pm$1.21 | 28.99$\pm$1.16 | 15.79$\pm$1.61 | 16.06$\pm$2.02 |
| SeqRNN (h=256) | 30.94$\pm$0.41 | 31.25$\pm$0.72 | 13.07$\pm$1.57 | 13.24$\pm$1.25 |
| SeqLSTM (h=128) | 24.96$\pm$0.46 | 25.78$\pm$0.43 | 10.13$\pm$1.24 | 10.39$\pm$1.50 |
| SeqLSTM (h=256) | 31.45$\pm$0.01 | 31.62$\pm$0.20 | 11.76$\pm$0.08 | 12.22$\pm$0.21 |
| StructMLP | 24.40$\pm$1.63 | 24.22$\pm$1.28 | 16.79$\pm$4.01 | 16.40$\pm$3.28 |
| StructGNN | 27.64$\pm$3.31 | 27.96$\pm$3.08 | 24.35$\pm$3.45 | 22.76$\pm$3.19 |
| GraphTrans | 23.81$\pm$2.57 | 22.21$\pm$2.98 | 17.32$\pm$5.28 | 17.04$\pm$5.36 |
| PiFold | 22.55$\pm$4.13 | 23.78$\pm$6.52 | 16.08$\pm$2.34 | 16.20$\pm$3.49 |
| MCTS-RNA | 31.74$\pm$0.07 | 32.06$\pm$1.87 | 23.82$\pm$4.60 | 24.12$\pm$3.47 |
| LEARNA | 31.92$\pm$2.37 | 30.94$\pm$4.15 | 24.02$\pm$3.73 | 22.75$\pm$1.17 |
| aRNAque | 30.01$\pm$3.26 | 31.07$\pm$2.32 | 22.84$\pm$1.70 | 23.30$\pm$1.65 |
| eM2dRNAs | 33.34$\pm$1.02 | 37.10$\pm$3.24 | 24.80$\pm$3.88 | 26.91$\pm$2.32 |
| RDesign | **56.12**$\pm$1.03 | **50.12**$\pm$1.07 | **53.27**$\pm$1.28 | **49.24**$\pm$1.07 |

## 4.3 ABLATION STUDY

We conducted an ablation study of RDesign and presented the results in Table 5. Firstly, we replaced our tertiary structure modeling approach with the classical modeling from protein design, which led to a significant decrease in performance. Secondly, removing the hierarchical representation learning also resulted in a performance drop, indicating its importance. Replacing the hyperspherical space with Euclidean space led to a substantial reduction in performance, indicating its impact on data-efficient learning. In contrast, removing the secondary structure constraints provided a relatively small decrease in performance because RDesign itself could accurately generate RNA sequences. Additionally, we further tested the capability that our designed sequences could fold into desired tertiary structures, which is the final aim of the RNA sequences design problem. However, due to the lack of reliable RNA tertiary structure prediction tools, we are only able to conduct this experiment in a qualitative way. Results of three example structures from each length category have been reported and analyzed in Appendix 5.

Table 5: The ablation study of our model on three datasets.

| Method | Recovery (%) ↑ | | | Macro F1 (×100) ↑ | | |
| --- | --- | --- | --- | --- | --- | --- |
| | Ours | Rfam | RNA-Puzzles | Ours | Rfam | RNA-Puzzles |
| RDesign | 41.53 | 56.12 | 50.12 | 40.89 | 53.27 | 49.24 |
| w/o our modeling | 36.45 | 53.19 | 44.93 | 36.33 | 48.95 | 43.88 |
| w/o reprsentation learning | 37.12 | 52.17 | 46.88 | 36.52 | 49.22 | 46.36 |
| w/o hyperspherical space | 30.69 | 36.33 | 36.00 | 30.67 | 30.34 | 33.57 |
| w/o secondary structure | 38.55 | 54.83 | 47.69 | 38.77 | 52.85 | 47.62 |

## 5 EVALUATION THE CAPABILITY OF FOLDING FOR DESIGNED SEQUENCES

We used RhoFold (Shen et al., 2022) to predict the structures of RNA sequences designed by RDesign. Figure 5 shows three visualization examples: (a) a short sequence reconstructed by RDesign; (b) a long sequence that was designed with a similar structure and low structure deviation; (c) a complicated sequence that was designed with a similar structure but failed to achieve low structure deviation. These visualization examples demonstrate the effectiveness of our RDesign model in designing RNA sequences with structures similar to the target structure.

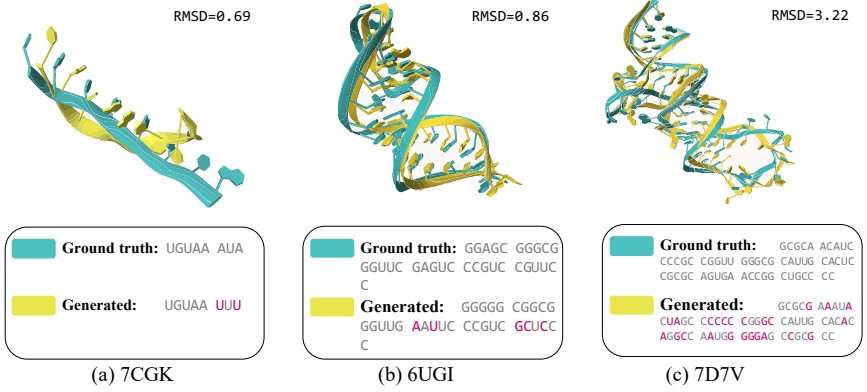

Figure 5: Visualization of RDesign's designed examples.

## 6 CONCLUSION AND LIMITATIONS

In this work, we investigate the challenging task of designing RNA tertiary structures. We compile a benchmark dataset to systematically assess the performance of various computational models on this task. While existing protein design methods cannot be directly applied, we propose a hierarchical data-efficient representation learning framework. Our framework explicitly captures the intrinsic relationships within the data while constraining the limited data to a restricted hyperspherical space. We also introduce a secondary structure constraining strategy to leverage extra structural information. Extensive experiments demonstrate the effectiveness of our proposed RDesign model. We hope this work provides a new perspective on tertiary structure-based RNA design. A limitation is that our method is currently limited to in silico design we leave wet-lab validation to future work.

## 7 ACKNOWLEDGEMENTS

We thank the anonymous reviewers for their constructive and helpful reviews. This work was supported by National Science and Technology Major Project (No. 2022ZD0115101), National Natural Science Foundation of China Project (No. U21A20427), the Center of Synthetic Biology and Integrated Bioengineering of Westlake University and Integrated Bioengineering of Westlake University Project (No. WU2022A009) and the Westlake University Industries of the Future Research Funding Project (No. WU2023C019).

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

## A EXPERIMENTAL SETTING

**Datasets** We evaluate our method on two tasks using three datasets: (i) We train and assess performance on our proposed RNA structure benchmark dataset which aggregates and cleans data from RNAsolo (Adamczyk et al., 2022b) and the Protein Data Bank (PDB) (Bank, 1971; Berman et al., 2000). The benchmark dataset consists of 2218 RNA tertiary structures, which are divided into training (1774 structures), testing (223 structures), and validation (221 structures) sets based on their structural similarity. The RNA sequence length is highly consistent within each subset, minimizing negative effects from distribution shifts. A detailed illustration of the creation and features of our dataset is shown in Appendix B. (ii) To test the generalization ability, we apply pre-trained models to the Rfam (Gardner et al., 2009; Nawrocki et al., 2015) and RNAPuzzle (Miao et al., 2020) datasets that contain non-overlapping structures. Rfam and RNAPuzzle are common datasets that contain novel RNA structures, which have 105 and 17 tertiary structures, respectively. The detailed description is in Appendix C.

**Metrics** Previous protein design approaches usually use sequence perplexity and recovery metrics to evaluate the performance (Li et al., 2014; Ingraham et al., 2019; Wang et al., 2018; Jing et al., 2020; Tan et al., 2022; Gao et al., 2022a; O'Connell et al., 2018). However, since RNA sequences only contain four letters, the perplexity metric traditionally used in natural language processing may not accurately reflect nucleotide-level prediction accuracy. Therefore, we use the recovery score to assess nucleotide-level accuracy by measuring the proportion of target RNA sequences that our model regenerates correctly. We also introduce the Macro-F1 score to evaluate performance by conceptualizing the RNA design task as a multi-class classification problem. The detailed illustrations of recovery and Macro-F1 scores are provided in Appendix E. We conducted each experiment three times with different random seeds and reported the mean and standard deviation values.

**Implementation Details** We trained the model for 200 epochs using the Adam optimizer with a learning rate of 0.001. The batch size was set as 64. The model was implemented based on the standard PyTorch Geometric (Fey & Lenssen, 2019) library using the PyTorch 1.11.0 library. We ran the models on Intel(R) Xeon(R) Gold 6240R CPU @ 2.40GHz CPU and NVIDIA A100 GPU.

The model's encoder and decoder each had three layers. With a dropout rate of 0.1, it considered 30 nearest neighbors and a vocabulary size matching RNA's four alphabets. For the baseline models, the loss function is the negative log-likelihood loss $\frac{1}{N} \sum_{i=1}^{N} - \log p(y_i)$, where $p(y)$ denotes the predicted probability of the true class. Notably, $p(y)$ are presented in logarithmic form. With the exception of SeqRNN (h=256) and SeqLSTM (h=256), which feature a hidden size of 256, the remaining models were configured with a hidden size of 128. In our study, we have repurposed StructGNN and GraphTrans, initially designed for protein design as described in Ingraham et al.'s work (Ingraham et al., 2019). These models have been adapted for RNA design, harnessing the capabilities of graph neural networks and the transformer architecture, respectively.

**Baseline Models** We explored six baseline models that could be categorized into four classes based on the level of structural information utilized: (i) Sequence-based models (SeqRNN and SeqLSTM) that do not utilize any structural information and could be viewed as the performance reference for RNA design; (ii) A tertiary structure-based model (StructMLP) that exploits structural features while ignoring the graph topological structure; (iii) Tertiary structure-based models (StructGNN and GraphTrans) (Ingraham et al., 2019) and PiFold (Gao et al., 2022b) that incorporate the graph topological structure; (iv) several state-of-the-art RNA sequence design models based on secondary structure (MCTS-RNA (Yang et al., 2017), LEARNA (Runge et al., 2018), eM2dRNAs (Rubio-Largo et al., 2023), aRNAque (Merleau & Smerlak, 2022)).

## B BENCHMARK DATASET DETAILS

### B.1 OVERVIEW

We meticulously gathered RNA tertiary structure data from two principal sources: RNAsolo (Adamczyk et al., 2022a) and the Protein Data Bank (PDB) (Berman et al., 2000). RNAsolo is an RNA 3D structure database that provides free online access for downloading. After the initial collection, we cleaned the structures by removing non-RNA data. We then annotated and assigned them to

equivalent classes defined by pairwise analysis of structural redundancy. We refined the abnormal samples that had inconsistent sequences and structures according to the PDB database.

Regarding the sequence length distribution, we provided a quantitative analysis shown in Table 6. From the original RNA solo dataset, $95.79\%$ of sequences are below 500 nucleotides in length, with sequences exceeding this length being sporadically dispersed between 500 and 4000 nucleotides.

Table 6: Sequence length distribution in RNA solo dataset.

| Range | Number of Sequences | Percentage (Cumulative) |
|---|---|---|
| 0-50 | 1875 | 69.81% |
| 50-100 | 483 | 87.79% |
| 100-500 | 215 | 95.79% |
| 500-1000 | 22 | 96.61% |
| 1000-2000 | 53 | 98.59% |
| 2000-3000 | 24 | 99.48% |
| 3000-4000 | 14 | 100.00% |

Incorporating such extremely long sequences would result in the selected features being overwhelmed by superfluous padding, thereby significantly escalating the computational resources needed. Therefore, we filtered out those with more than 500 nucleotides. We provide a runtime and memory usage analysis in Table 7.

Table 7: Runtime and memory usage evaluation of RDesign.

| Sequence Length | Memory (MB) | Runtime (Second) |
|---|---|---|
| 500 | 7705 (1.00 $\times$) | 7.5 (1.00 $\times$) |
| 2000 | 16499 (2.14 $\times$) | 33.6 (4.48 $\times$) |
| 4000 | 48913 (6.35 $\times$) | 64.0 (8.53 $\times$) |

After filtering out the sequences with more than 500 nucleotides, we collected a total of 2218 high-quality representative RNA structures, each with a resolution of less than 4.0 Å.

## B.2 DATA CLUSTERING

We used the TM-score (Zhang & Skolnick, 2005) to assess structural similarity among RNA structures. To group samples with similar structures, we employed a graph-based clustering approach. Each structure is represented as a node, and the TM-Score between every pair of structures is computed. If the TM-Score for a pair exceeds 0.45, an edge is drawn between them, indicating their high similarity. After processing all pairs in this manner, clusters are identified as connected components within the graph. From our dataset of 2218 RNA structures, we yielded 987 distinct clusters.

## B.3 DATA SPLITTING

How to split biological datasets in a way that avoids information leakage and provides consistent evaluation is an important topic in AI for science. To achieve this, we split the collected data based on two principles: (i) avoiding similar structures in different sets and (ii) maintaining similar length distributions across sets. After obtaining 987 clusters, we added each to the training, validation, and testing datasets, maintaining vigilance to prevent similar structures from appearing in divergent sets. Specifically, we calculated the average length of all sequences and sequentially allocated clusters to the respective train/validation/test sets, managing to align each cluster closely with the global average length. Consequently, we maintained a consistent sequence length distribution across all sets, as depicted in Figure 6.

In this way, we allocated entire clusters—rather than individual samples—to specific sets, eliminating the possibility of data leakage between sets. Of the 987 total clusters, 838 were designated to the

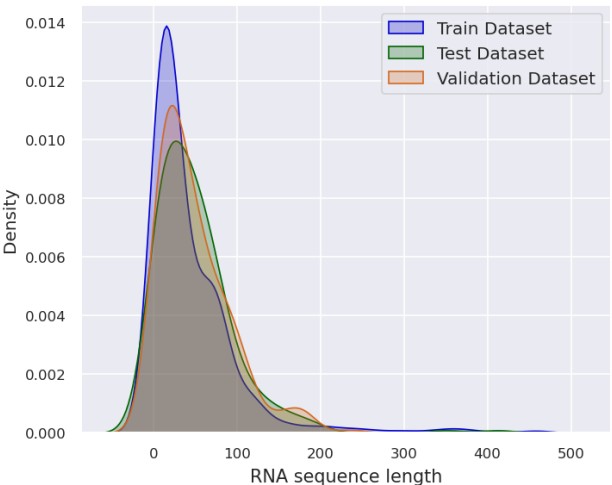

Figure 6: Evaluation on sequence length distribution in the train, test, and validation set.

training set, 68 to the validation set, and 81 to the test set. The cluster size distribution in each set is shown in Table 8. It is worth noting that clusters containing a large number of samples (size $> 30$) are placed in the training set rather than the validation and test sets to prevent bias from dominant clusters when evaluating model performance.

Table 8: Size distribution of clusters in each dataset

| Dataset | Size $\leq 2$ | $3 \leq$ Size $\leq 10$ | $10 <$ Size $\leq 30$ | Size $> 30$ |
|---|---|---|---|---|
| Train | 810 | 25 | 0 | 3 |
| Validation | 41 | 26 | 1 | 0 |
| Test | 53 | 26 | 2 | 0 |

Finally, we partitioned the original dataset into training (1774 structures), testing (223 structures), and validation (221 structures) sets based on structural similarity and RNA sequence length. The evaluations of the baseline models were conducted on this split to ensure a fair comparison.

### B.4 DATA AUGMENTATION

For the purpose of sample-level representation learning, we employed a method to generate pseudo RNA structures by introducing slight perturbations to the coordinates. This approach takes into account the possibility of minor errors in experimental data. Initially, we obtained the coordinates of each atom within the RNA chains and introduced Gaussian noise, with the magnitude of noise determined by the scale of each coordinate value.

Subsequently, we assessed the structural fidelity of the generated data by calculating two key metrics: the Root Mean Square Deviation (RMSD) and the TM-score in comparison to the ground truth structure. We excluded any generated data with an RMSD exceeding $1.0$ Å or a TM-score lower than $0.8$. This stringent criterion was applied to ensure that the augmented samples maintained structural characteristics closely resembling those of the original data.

### C RFAM AND RNA-PUZZLES DATASET DETAILS

#### C.1 RFAM DATABASE

The Rfam database (Kalvari et al., 2021) is a collection of RNA families, each represented by a multiple sequence alignment, a consensus secondary structure, and covariance models that describe

the consensus structure and base-pairing constraints of the RNA family. In the 14.7 release, 122 usable RNA sequences are aligned with their 3D structure stored in PDB format.

## C.2 RNA-PUZZLES DATASET

The RNA-puzzles dataset (Miao et al., 2020) is a public dataset that consists of a series of RNA structure prediction challenges, designed to evaluate and improve the accuracy of computational methods for predicting the structure of RNA molecules. Each RNA-puzzles challenge provides a set of experimental data, such as X-ray crystallography measurements stored in PDB format, along with a target sequence and structure for the RNA molecule. It has been widely used to benchmark and compare different computational methods for RNA structure prediction.

## D   LOCAL COORDINATE SYSTEMS UTILIZED IN OUR PAPER

Generally, the conventional modeling mentioned is the common protein tertiary structural modeling approach from (Ingraham et al., 2019). It considers dihedral angles as node features and uses $C\alpha$ atoms to build the local coordinate system. While it is sufficient for extracting features from protein structures, they may not be suitable for the more complex RNA structures. We propose a comprehensive approach to modeling RNA tertiary structures that are specifically designed for RNA.

Our baseline models StructGNN and GraphTrans are from (Ingraham et al., 2019), which uses the conventional modeling approach. We adjusted this protein modeling for RNA by replacing its three dihedral angles with RNA's six and using the P atom instead of the $C\alpha$ atom.

We construct a local coordinate system for each nucleotide $i$. This system is defined as:

$$Q_i = [\mathbf{b}_i, \mathbf{n}_i, \mathbf{b}_i \times \mathbf{n}_i], \tag{10}$$

where $\mathbf{b}_i$ is the negative bisector of angles between the rays of contiguous coordinates $(\boldsymbol{x}_{i-1,\mathrm{P}}, \boldsymbol{x}_{i,\mathrm{P}})$ and $(\boldsymbol{x}_{i+1,\mathrm{P}}, \boldsymbol{x}_{i,\mathrm{P}})$, and $\mathbf{n}_i$ is a unit vector normal to that plane. Formally, $\mathbf{b}_i$ and $\mathbf{n}_i$ are:

$$\mathbf{u}_i = \frac{\mathbf{x}_i - \mathbf{x}_{i-1}}{\|\mathbf{x}_i - \mathbf{x}_{i-1}\|}, \mathbf{b}_i = \frac{\mathbf{u}_i - \mathbf{u}_{i+1}}{\|\mathbf{u}_i - \mathbf{u}_{i+1}\|}, \mathbf{n}_i = \frac{\mathbf{u}_i \times \mathbf{u}_{i+1}}{\|\mathbf{u}_i \times \mathbf{u}_{i+1}\|}. \tag{11}$$

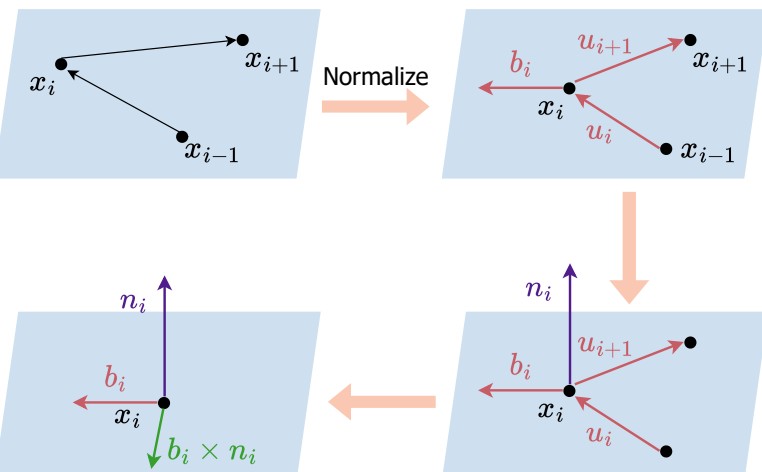

Figure 7: Procedure of building the local coordinates.

As illustrated in Figure 7, constructing the local coordinate system involves calculating the bond vectors between successive P atoms and using these to determine the bisector $\mathbf{b}_i$ and normal $\mathbf{n}_i$. Since this coordinate system is defined relative to a fixed set of atoms within each nucleotide, it is invariant to overall rotations and translations of the RNA structure.

# E EVALUATION METRICS

## E.1 RECOVERY

To rigorously assess the nucleotide-level prediction accuracy of an RNA sequence model, we adopt the following recovery metric:

$$\text{Recovery}(\mathcal{S}^N, \mathcal{X}^N) = \frac{1}{N} \sum_{i=1}^{N} \mathbb{1}\Big[\mathcal{S}_i^N = \arg\max \mathcal{P}(\mathcal{S}_i^N | \mathcal{X})\Big], \tag{12}$$

where $\mathcal{P}(S_i^N | \mathcal{X}_i, \mathcal{X}_{<i})$ represents the probability of predicting nucleotide $S_i^N$ at position $i$ given the input structure $\mathcal{X}$, and $\mathbb{1}[\odot]$ is the indicator function. This recovery metric evaluates the fraction of nucleotides in the RNA sequence that are correctly predicted as the most probable by the model, averaged over all $N$ positions. It thus rigorously assesses the model's ability to accurately predict the sequence at the nucleotide level based on the inputs. This metric provides a nuanced evaluation of the predictive accuracy achieved by the RNA sequence model at the nucleotide resolution.

## E.2 MACRO-F1 SCORE

We introduce the Macro-F1 score to evaluate performance by conceptualizing the RNA design task as a multi-class classification problem and adopt the Macro-F1 score metric to comprehensively evaluate the model's performance on different classes. It is calculated as the average of the F1 scores for each class of letters.

$$\text{Macro-F1} = \frac{1}{|C|} \sum_{c \in \{A,U,C,G\}}^{C} \text{F1}_c, \tag{13}$$

where $|C|$ is the number of letters and $F1_c$ is the F1 score for letter $c$. The F1 score is the harmonic mean of precision and recall. It considers both the proportion of correctly predicted instances of letter $c$ and the proportion of instances predicted as letter $c$ that belong to letter $c$.

$$\text{F1}_c = 2 \times \frac{\text{Precision}_c \times \text{Recall}_c}{\text{Precision}_c + \text{Recall}_c}. \tag{14}$$

Therefore, the Macro-F1 score comprehensively reflects the model's prediction performance on different letters by evaluating the F1 score for each letter. Compared to simply calculating the average precision or recall over all letters, it can more comprehensively evaluate the model's ability to identify instances of each letter, especially in cases where there is an imbalance between the number of instances of different letters. It examines the proportion of correctly classified instances at the letter level while balancing precision and recall.

## E.3 PERPLEXITY

Perplexity is a common metric used in various fields including natural language processing and bioinformatics. In the context of RNA design, perplexity can be used as a measure to evaluate the uncertainty of RNA sequences, reflecting how well a model can predict a given RNA sequence.

$$\text{Perplexity}(\mathcal{S}^N, \mathcal{X}^N) = \exp(-\frac{1}{N} \sum_{i=1}^{N} \mathcal{S}_i^N \log p(\mathcal{S}_i^N | \mathcal{X}^N)), \tag{15}$$

Where $p(\mathcal{S}_i^N | \mathcal{X}^N)$ is the output probability from the model.

While perplexity is a crucial metric in natural language processing (NLP) and protein design, its suitability is significantly less in RNA design due to the inherent simplicity and structural constraints of RNA sequences. RNA sequences, with only four letters A, U, C, G, lack the compositional complexity found in natural languages, making perplexity less discriminative and revealing as a model evaluation metric in this context. Considering these limitations, we chose not to prioritize this metric in our main text. We show the results in Table 9. It can be seen that the outcomes are not aligned with other metrics, and notably, sequence-based methods consistently outperform structure-based methods, which is a phenomenon contrary to common sense.

Table 9: The perplexity metric on the benchmark dataset.

| Method | Perplexity (%)↓ | | | |
|---|---|---|---|---|
| | Short | Medium | Long | All |
| SeqRNN (h=128) | 4.02±0.02 | 4.01±0.01 | 4.00±0.01 | 4.01±0.01 |
| SeqRNN (h=256) | 4.00±0.01 | 3.99±0.00 | 3.99±0.00 | 3.99±0.01 |
| SeqLSTM (h=128) | 4.00±0.01 | 4.00±0.00 | 3.99±0.01 | 4.00±0.01 |
| SeqLSTM (h=256) | 4.00±0.00 | 3.99±0.00 | 3.98±0.00 | 3.99±0.00 |
| StructMLP | 10.83±1.32 | 7.78±2.41 | 7.93±2.74 | 8.51±2.35 |
| StructGNN | 4.87±0.36 | 4.87±0.48 | 5.12±0.43 | 4.95±0.43 |
| GraphTrans | 5.79±1.65 | 6.32±2.23 | 6.56±2.26 | 6.25±2.07 |
| RDesign | 4.28±0.14 | **3.98**±0.10 | **3.97**±0.04 | 4.10±0.09 |

## F HYPERSPHERICAL SPACE OF REPRESENTATION LEARNING

In contrastive representation learning, mapping features to hyperspherical space is a technique used to better represent similarity. In this supervised learning algorithm, the similarity between two samples is represented as the distance or similarity between their feature vectors. To better represent this similarity, the features are mapped to hyperspherical space. Specifically, the feature vector $h$ is mapped to hyperspherical space by dividing it by its norm $||h||$, resulting in a unit vector $\frac{h}{||h||}$. This unit vector is the representation in hyperspherical space. The projection head comprises an MLP and feature normalization. The advantage of this method is that it normalizes feature vectors to the same length, avoiding the influence of scale changes in the feature space on similarity measurement. Additionally, since hyperspherical space is a convex space, similarity measurement in this space is convenient and efficient.

Using a hypersphere for embedding data points is advantageous because it allows each data point to cover a larger expressive space compared to confining all points within a small region of Euclidean space. This enables the model to maximize the expressive power and learnability of each individual data point within the limited dataset. Conversely, if data distributions are not controlled, the representations may collapse and cluster in a small portion of the space, limiting their individual expressiveness. The hypersphere provides a structured way to distribute data representations evenly. This uniform assignment of limited data to hypersphere space is thus crucial for ensuring every data point can be maximally informative during contrastive representation learning. The design choice stems directly from the need to learn efficiently from scarce RNA structure data.

## G MODEL DETAILS

### G.1 RDESIGN MODEL

Here we present two levels of the system paradigm of our model. The brief pipeline is shown in Figure 8, which serves as a comparison with the baselines proposed in our paper. Another detailed pipeline is presented as Figure 9, which draws a comprehensive view of our model, indicating the function of each part, and illustrating the formulation of the loss function.

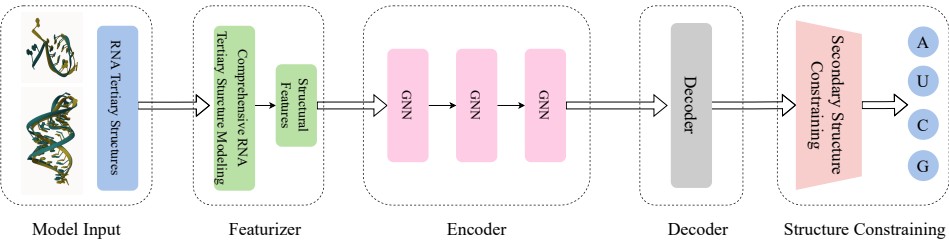

Figure 8: The pipeline of RDesign model.

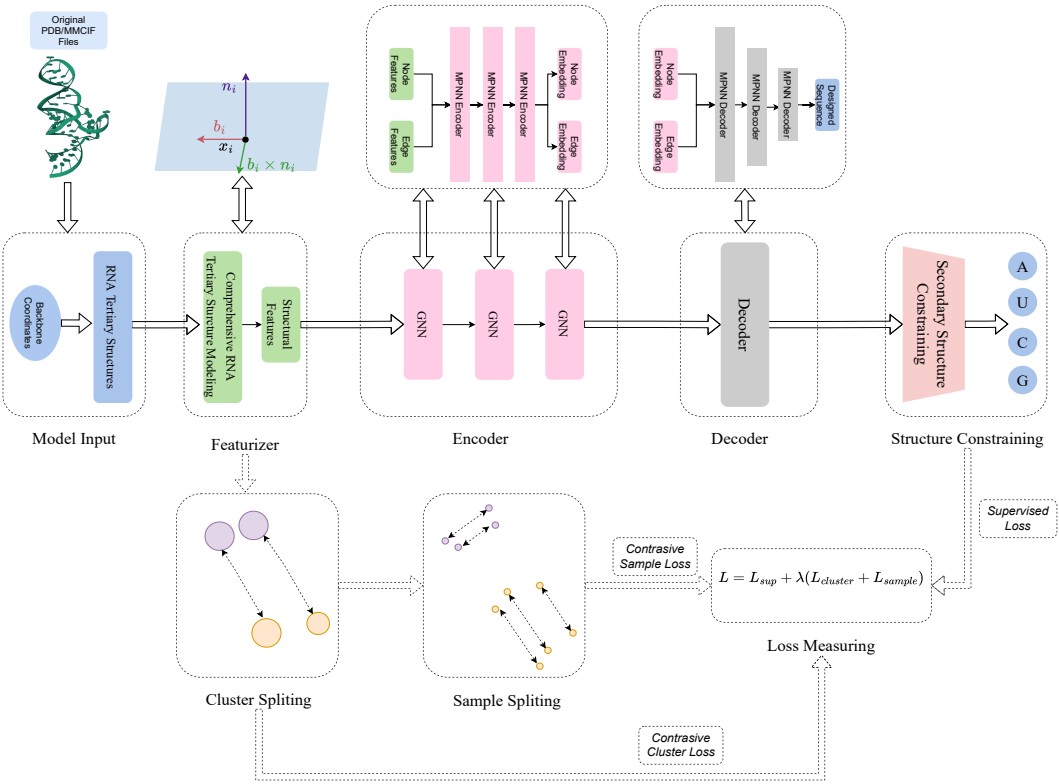

Figure 9: Detailed system pipeline of our model.

## G.2 BASELINE MODELS

In this work, we employed five baseline models to perform as the reference to our RDesign model. For each of them, we provide a brief introduction and summarize the pipeline in the figure below. In the experiment, both the encoder and decoder contain three layers by default.

**SeqRNN** SeqRNN is a context-aware recurrent neural network model that is suitable for sequence generation tasks. Based on recurrent networks, it can utilize its memory to process the input sequences and learn the distribution of the sequences by adjusting the weight coefficients. The output of SeqRNN is a sequence with the same length as the inputs. The pipeline is shown in Figure 10.

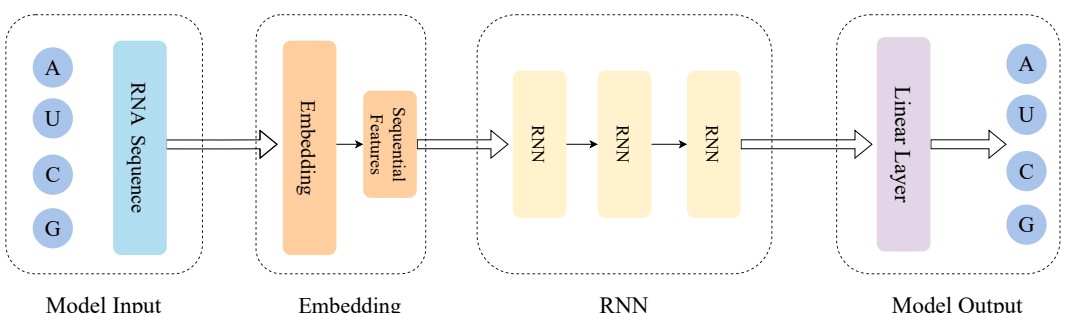

Figure 10: The pipeline of SeqRNN model.

**SeqLSTM** Similar to SeqRNN, SeqLSTM is an LSTM-based context-aware suitable for sequence generation tasks. It can extract the sequence order features into embeddings. Then the output sequence could be obtained by passing the embeddings through a linear layer. The pipeline of the model is shown in Figure 11.

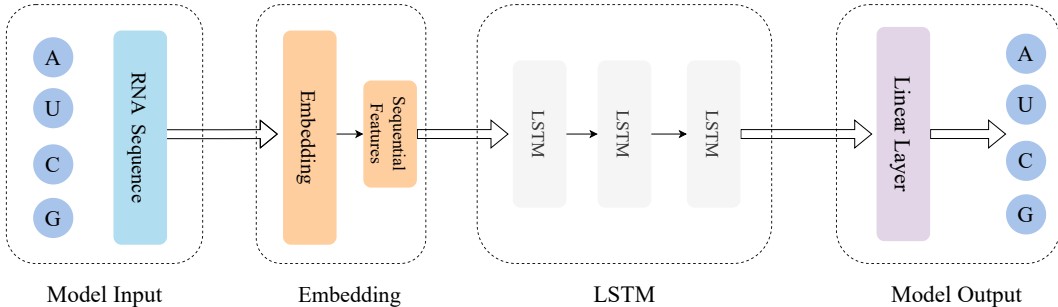

Figure 11: The pipeline of SeqLSTM model.

**StructMLP** StructMLP represents a structure-based RNA sequence prediction model. This approach harnesses the structural characteristics of RNA molecules through a Multilayer Perceptron (MLP) architecture, allowing for the structural insights to produce sequence predictions. The schematic representation of the model's workflow is illustrated in Figure 12.

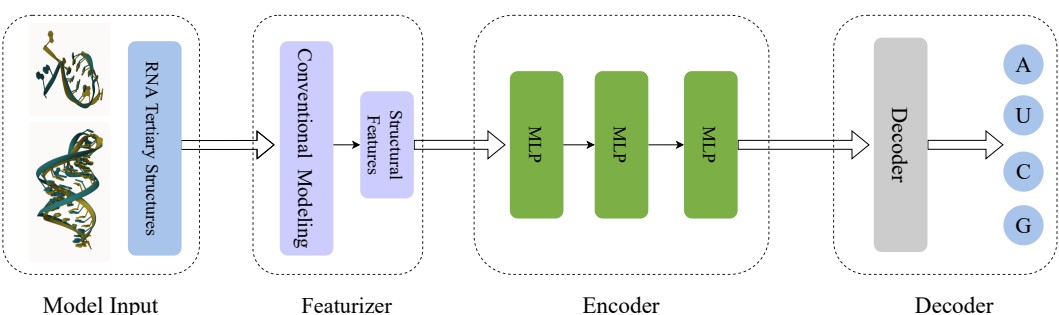

Figure 12: The pipeline of StructMLP model.

**StructGNN** In contrast to StructMLP, StructGNN employs a Graph Neural Network (GNN) as its feature encoder. This choice equips StructGNN with the capacity to capture and learn the intricate graph structures inherent in RNA tertiary structures, a capability beyond the reach of StructMLP. The model's workflow is visually depicted in Figure 13.

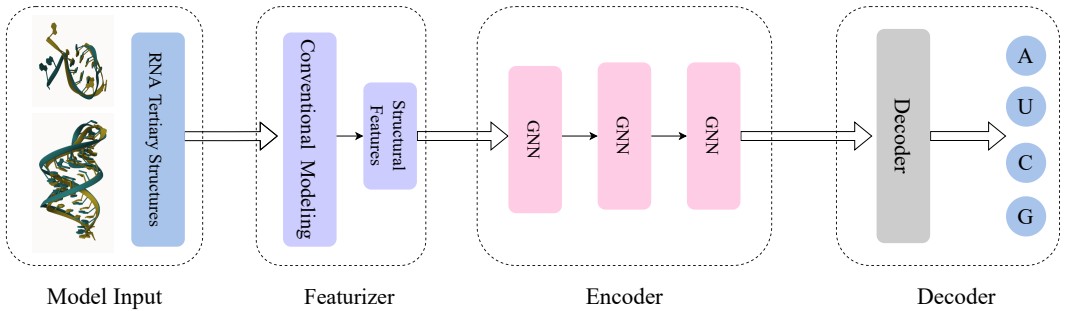

Figure 13: The pipeline of StructGNN model.

**GraphTrans** The primary distinction between GraphTrans and StructGNN lies in the feature encoder employed. GraphTrans leverages the Transformer architecture for this purpose. To illustrate the model's workflow, please refer to Figure 14.

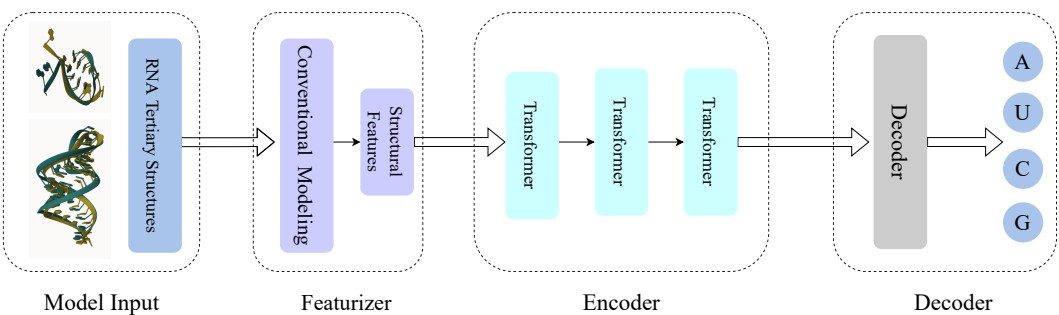

Figure 14: The pipeline of GraphTrans model.

# H EVALUATION ON EXTERNAL DATASETS

In the main text, we used the pretrained model trained on the entire training set to evaluate models on the external datasets Rfam and RNA-Puzzles. We show the structural similarity distribution between our training data and the three test sets in Figure 15. This distribution demonstrates that the majority of structural similarities are concentrated around a TM-score of 0.1, which suggests low structural similarity. In our benchmark dataset, the structural similarity between all test data and any training sample remains below 0.45. However, in the Rfam and RNA-Puzzles datasets, there are still a few instances of similar samples. We directly evaluated the models pretrained on the complete training set on these two datasets, as this approach closely mirrors real-world application scenarios.

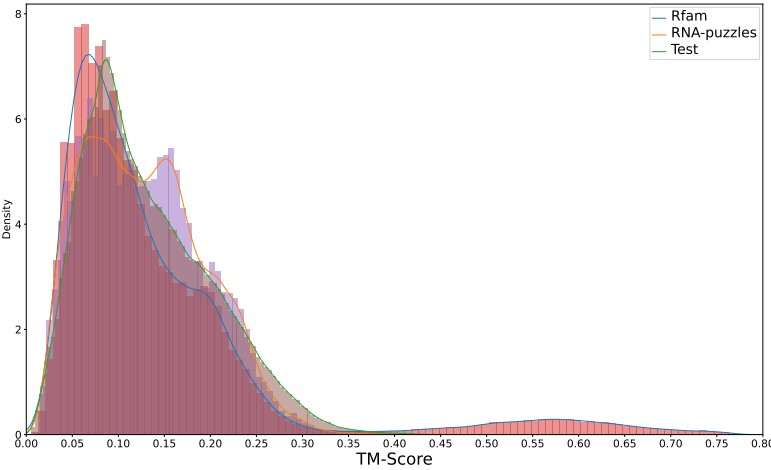

Figure 15: The pairwise structural similarity between our training set and the three test sets.

To evaluate the model's generalization ability on dissimilar samples, we implemented a filtering process on the training set. While evaluating models on the Rfam dataset, we removed all samples from the training set that exhibited a structural similarity exceeding 0.45 with any sample in Rfam. Likewise, during the evaluation on the RNA-Puzzles dataset, we excluded all samples from the training set that possessed structural similarity scores surpassing 0.45 with any sample in RNA-Puzzles. The distributions of pairwise structural similarity between the training set and three test sets are visualized in Figure 16.

The results on the Rfam and RNA-Puzzles with filtered training sets are shown in Table 10. It can be seen that RDesign still outperforms other baseline models.

# I COMPARISON ON AUROC AND AUPRC METRICS

We conduct comparisons with baseline models using the AUROC and AUPRC metrics. For our benchmark test set, the results for the AUROC and AUPRC metrics are presented in Tables 11 and

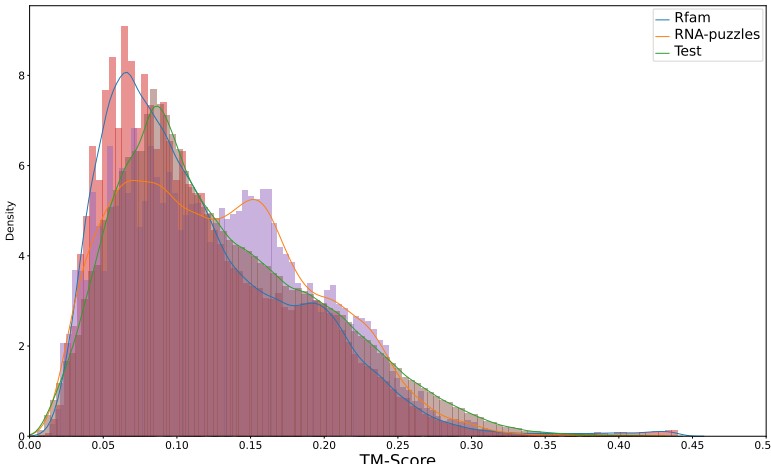

Figure 16: The pairwise structural similarity between our training set and the three test sets after removing similar structures in Rfam and RNA-Puzzles.

Table 10: The recovery and Macro-F1 scores on the Rfam and RNA-Puzzles datasets with filtered training sets.

| Method | Recovery (%) ↑ | | Macro F1 (×100) ↑ | |
|---|---|---|---|---|
| | Rfam | RNA-Puzzles | Rfam | RNA-Puzzles |
| SeqRNN (h=128) | 31.05±0.51 | 31.51±0.05 | 11.92±0.17 | 12.11±0.03 |
| SeqRNN (h=256) | 31.04±0.50 | 31.53±0.04 | 11.93±0.16 | 12.12±0.02 |
| SeqLSTM (h=128) | 30.28±0.20 | 31.35±0.26 | 12.36±0.15 | 12.40±0.15 |
| SeqLSTM (h=256) | 31.45±0.08 | 31.79±0.44 | 11.76±0.09 | 12.07±0.00 |
| StructMLP | 26.77±3.38 | 27.06±3.81 | 16.22±2.43 | 16.72±2.53 |
| StructGNN | 20.81±1.42 | 20.68±0.70 | 14.54±1.11 | 12.70±2.60 |
| GraphTrans | 27.50±4.15 | 25.69±4.34 | 20.66±2.51 | 20.17±0.14 |
| RDesign | **41.28**±0.35 | **43.99**±1.67 | **38.76**±0.82 | **43.11**±0.38 |

12, respectively. Regarding the Rfam and RNA-Puzzles datasets, the AUROC and AUPRC results are displayed in Tables 13, 14, 15, and 16, respectively. RDesign consistently outperforms the other baseline models in these datasets.

Table 11: AUROC on our benchmark test set.

| Method | A | U | C | G |
|---|---|---|---|---|
| SeqRNN (h=128) | 49.40 ± 0.18 | 50.29 ± 0.58 | 51.01 ± 0.60 | 49.48 ± 1.21 |
| SeqRNN (h=256) | 51.91 ± 0.57 | 49.77 ± 0.51 | 51.58 ± 0.32 | 50.88 ± 0.41 |
| SeqLSTM (h=128) | 49.38 ± 0.30 | 50.04 ± 2.38 | 49.80 ± 2.03 | 49.16 ± 1.32 |
| SeqLSTM (h=256) | 50.06 ± 0.95 | 51.48 ± 0.77 | 47.65 ± 0.64 | 50.44 ± 0.68 |
| StructMLP | 48.82 ± 1.17 | 50.10 ± 3.08 | 51.26 ± 3.01 | 49.44 ± 2.60 |
| StructGNN | 53.17 ± 1.02 | 49.00 ± 2.58 | 47.92 ± 1.70 | 50.59 ± 1.42 |
| GraphTrans | 53.06 ± 0.44 | 49.19 ± 4.62 | 47.44 ± 3.14 | 52.08 ± 2.62 |
| PiFold | 49.09 ± 1.60 | 49.97 ± 2.70 | 50.51 ± 4.13 | 51.21 ± 0.74 |
| RDesign | **66.16** ± 0.27 | **63.22** ± 0.38 | **65.58** ± 0.41 | **67.31** ± 0.21 |

Table 12: AUPRC on our benchmark test set.

| Method | A | U | C | G |
|---|---|---|---|---|
| SeqRNN (h=128) | 24.61 ± 0.23 | 25.97 ± 0.44 | 23.36 ± 0.66 | 27.45 ± 0.34 |
| SeqRNN (h=256) | 26.90 ± 0.46 | 25.64 ± 0.34 | 23.83 ± 0.34 | 28.97 ± 0.24 |
| SeqLSTM (h=128) | 24.49 ± 0.23 | 25.56 ± 1.60 | 22.72 ± 1.19 | 27.55 ± 1.49 |
| SeqLSTM (h=256) | 24.78 ± 0.57 | 26.40 ± 0.25 | 21.79 ± 0.16 | 28.29 ± 0.65 |
| StructMLP | 24.72 ± 0.57 | 25.17 ± 1.83 | 23.47 ± 1.60 | 27.18 ± 1.30 |
| StructGNN | 27.18 ± 0.49 | 24.63 ± 1.68 | 22.14 ± 0.74 | 28.33 ± 0.84 |
| GraphTrans | 26.65 ± 0.25 | 24.88 ± 2.87 | 22.07 ± 1.84 | 28.66 ± 1.62 |
| PiFold | 24.30 ± 0.84 | 25.01 ± 1.78 | 23.41 ± 2.28 | 28.62 ± 0.61 |
| RDesign | **40.37** ± 0.08 | **34.60** ± 0.89 | **34.91** ± 0.89 | **45.17** ± 0.81 |

Table 13: AUROC on Rfam dataset.

| Method | A | U | C | G |
|---|---|---|---|---|
| SeqRNN (h=128) | 50.60 ± 1.14 | 50.06 ± 1.65 | 52.43 ± 1.04 | 49.33 ± 1.10 |
| SeqRNN (h=256) | 51.60 ± 0.71 | 50.11 ± 0.29 | 51.62 ± 0.67 | 52.32 ± 1.64 |
| SeqLSTM (h=128) | 49.24 ± 1.47 | 50.31 ± 1.52 | 50.59 ± 2.35 | 47.58 ± 1.56 |
| SeqLSTM (h=256) | 50.04 ± 1.36 | 50.01 ± 0.88 | 50.34 ± 0.85 | 50.00 ± 1.41 |
| StructMLP | 46.97 ± 2.20 | 52.09 ± 2.96 | 49.77 ± 2.80 | 47.56 ± 1.43 |
| StructGNN | 53.18 ± 0.85 | 50.93 ± 1.69 | 48.10 ± 2.37 | 51.58 ± 0.37 |
| GraphTrans | 51.38 ± 2.72 | 48.82 ± 1.57 | 47.51 ± 2.07 | 53.04 ± 1.72 |
| PiFold | 49.34 ± 2.91 | 48.86 ± 1.78 | 49.90 ± 3.07 | 51.57 ± 0.92 |
| RDesign | **67.89** ± 0.24 | **64.87** ± 0.25 | **66.42** ± 0.57 | **66.78** ± 0.06 |

Table 14: AUPRC on Rfam dataset.

| Method | A | U | C | G |
|---|---|---|---|---|
| SeqRNN (h=128) | 22.79 ± 0.66 | 21.10 ± 0.96 | 27.61 ± 0.47 | 31.32 ± 1.18 |
| SeqRNN (h=256) | 23.66 ± 0.40 | 20.81 ± 0.40 | 27.37 ± 0.67 | 33.87 ± 1.24 |
| SeqLSTM (h=128) | 22.11 ± 0.66 | 21.46 ± 1.45 | 26.71 ± 1.67 | 30.08 ± 1.37 |
| SeqLSTM (h=256) | 22.80 ± 0.75 | 20.86 ± 0.32 | 26.12 ± 0.37 | 31.61 ± 1.75 |
| StructMLP | 21.55 ± 1.37 | 22.31 ± 2.32 | 25.53 ± 1.39 | 29.07 ± 0.92 |
| StructGNN | 25.45 ± 0.67 | 21.37 ± 1.56 | 25.74 ± 1.49 | 31.99 ± 1.03 |
| GraphTrans | 24.09 ± 1.40 | 20.14 ± 1.23 | 25.16 ± 1.46 | 32.92 ± 1.24 |
| PiFold | 22.83 ± 1.53 | 20.31 ± 1.33 | 25.72 ± 1.70 | 32.63 ± 0.68 |
| RDesign | **40.40** ± 0.85 | **31.65** ± 0.15 | **42.36** ± 0.26 | **47.36** ± 0.24 |

Table 15: AUROC on RNA-Puzzles.

| Method | A | U | C | G |
|---|---|---|---|---|
| SeqRNN (h=128) | 49.25 ± 1.01 | 49.70 ± 0.31 | 50.80 ± 1.40 | 49.03 ± 0.87 |
| SeqRNN (h=256) | 49.29 ± 0.68 | 49.31 ± 1.45 | 50.69 ± 0.53 | 51.78 ± 0.78 |
| SeqLSTM (h=128) | 48.53 ± 0.81 | 50.39 ± 1.90 | 49.62 ± 1.61 | 47.57 ± 1.76 |
| SeqLSTM (h=256) | 50.63 ± 1.43 | 49.94 ± 1.15 | 48.44 ± 1.79 | 49.13 ± 2.33 |
| StructMLP | 44.98 ± 2.31 | 51.49 ± 3.67 | 51.11 ± 5.15 | 46.92 ± 1.91 |
| StructGNN | 53.46 ± 1.59 | 48.81 ± 2.88 | 46.97 ± 3.26 | 50.62 ± 1.35 |
| GraphTrans | 51.99 ± 2.02 | 47.48 ± 0.33 | 46.14 ± 1.91 | 51.84 ± 3.13 |
| PiFold | 49.94 ± 2.69 | 48.68 ± 3.67 | 49.55 ± 6.09 | 52.49 ± 0.57 |
| RDesign | **71.43** ± 0.77 | **70.69** ± 1.01 | **70.56** ± 0.78 | **70.58** ± 0.59 |

Table 16: AUPRC on RNA-Puzzles.

| Method | A | U | C | G |
|---|---|---|---|---|
| SeqRNN (h=128) | $22.15 \pm 0.53$ | $20.13 \pm 0.39$ | $25.90 \pm 0.87$ | $32.74 \pm 0.92$ |
| SeqRNN (h=256) | $22.50 \pm 0.15$ | $18.90 \pm 0.41$ | $26.10 \pm 0.65$ | $34.50 \pm 0.23$ |
| SeqLSTM (h=128) | $21.88 \pm 0.59$ | $19.74 \pm 1.28$ | $25.64 \pm 0.72$ | $32.00 \pm 1.75$ |
| SeqLSTM (h=256) | $22.91 \pm 1.20$ | $19.62 \pm 0.40$ | $25.00 \pm 0.73$ | $31.82 \pm 1.98$ |
| StructMLP | $20.32 \pm 1.06$ | $21.40 \pm 2.55$ | $26.48 \pm 3.27$ | $29.76 \pm 1.44$ |
| StructGNN | $25.85 \pm 2.14$ | $19.18 \pm 2.00$ | $24.12 \pm 1.41$ | $33.30 \pm 0.95$ |
| GraphTrans | $24.82 \pm 1.08$ | $18.53 \pm 0.19$ | $24.07 \pm 1.70$ | $33.08 \pm 3.34$ |
| PiFold | $23.58 \pm 1.34$ | $18.49 \pm 1.52$ | $25.44 \pm 3.50$ | $34.43 \pm 1.13$ |
| RDesign | $\mathbf{48.93} \pm 0.58$ | $\mathbf{37.74} \pm 1.87$ | $\mathbf{46.44} \pm 0.95$ | $\mathbf{53.22} \pm 0.12$ |

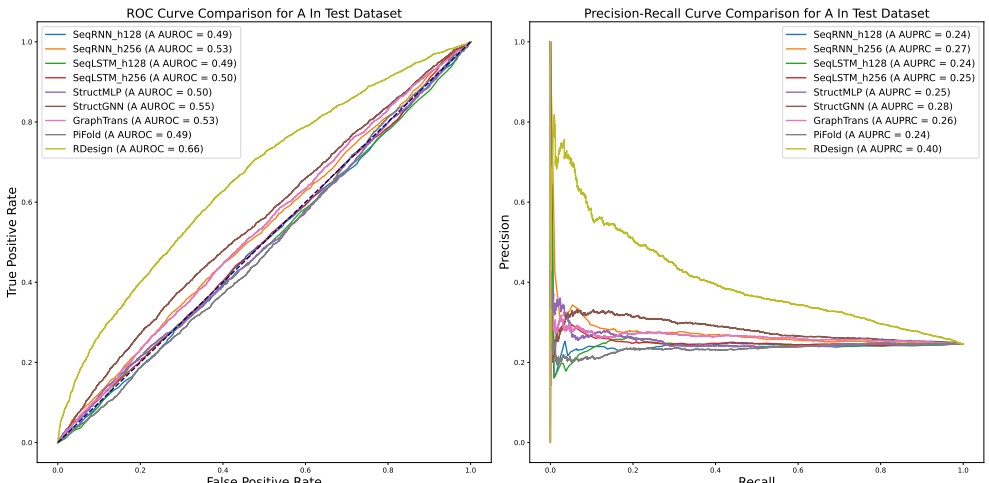

Figure 17: The ROC/PRC curve comparison on the base A of our benchmark test set.

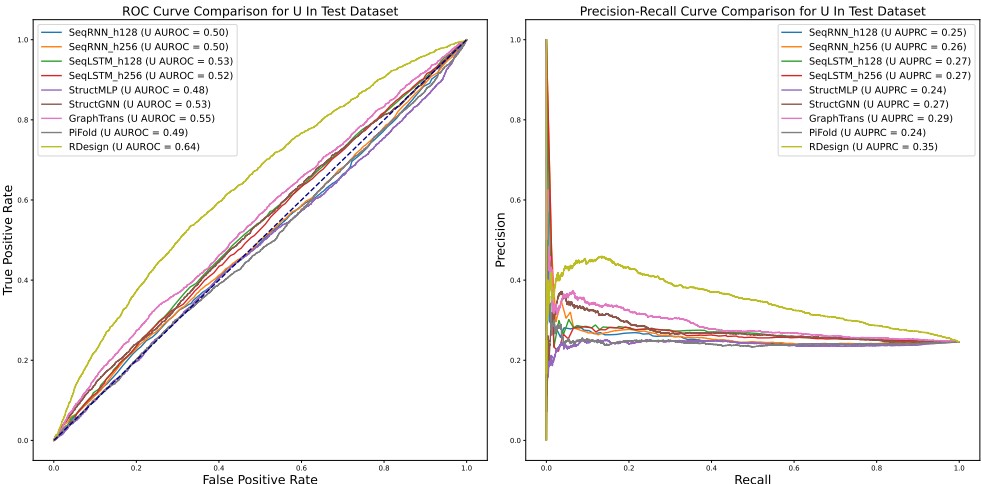

Figure 18: The ROC/PRC curve comparison on the base U of our benchmark test set.

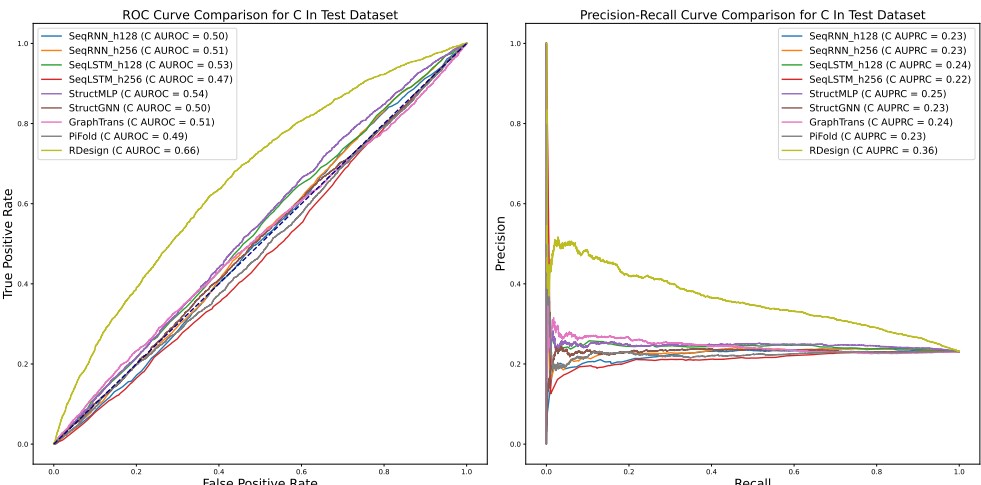

Figure 19: The ROC/PRC curve comparison on the base C of our benchmark test set.

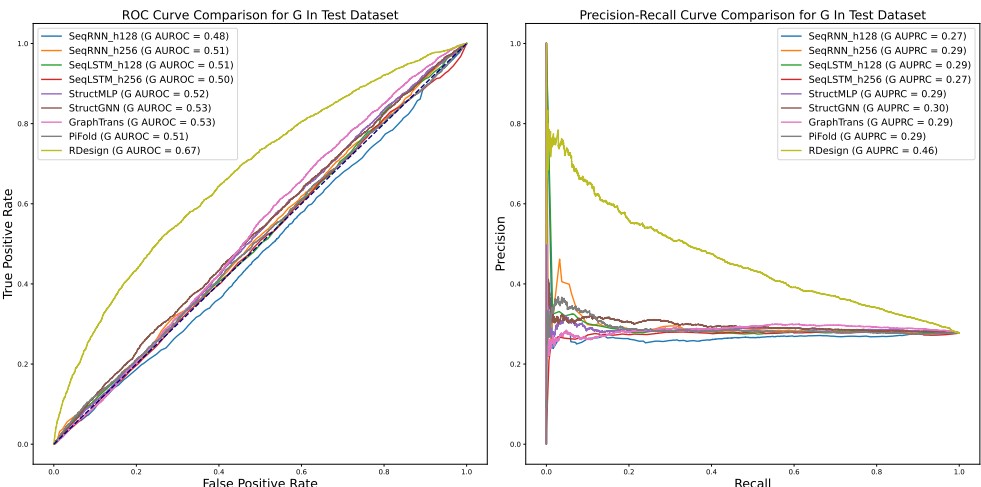

Figure 20: The ROC/PRC curve comparison on the base G of our benchmark test set.

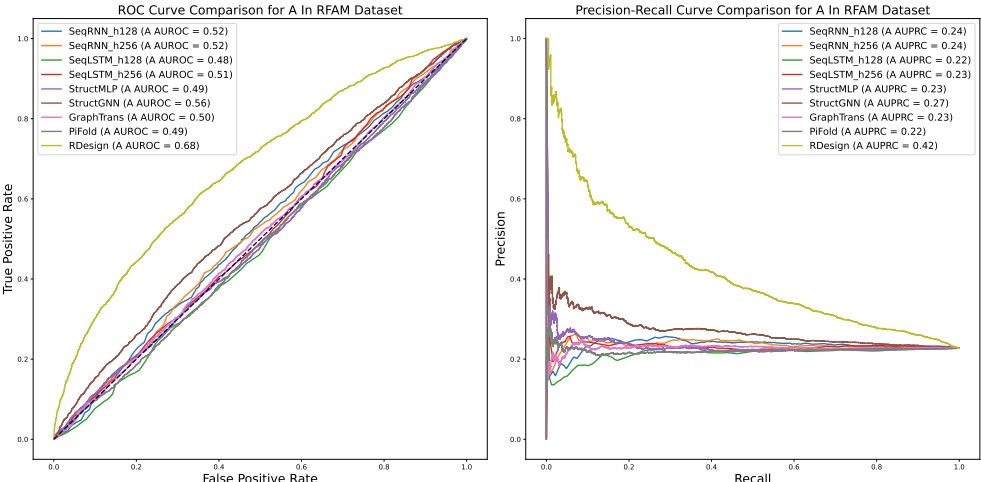

Figure 21: The ROC/PRC curve comparison on the base A of Rfam dataset.

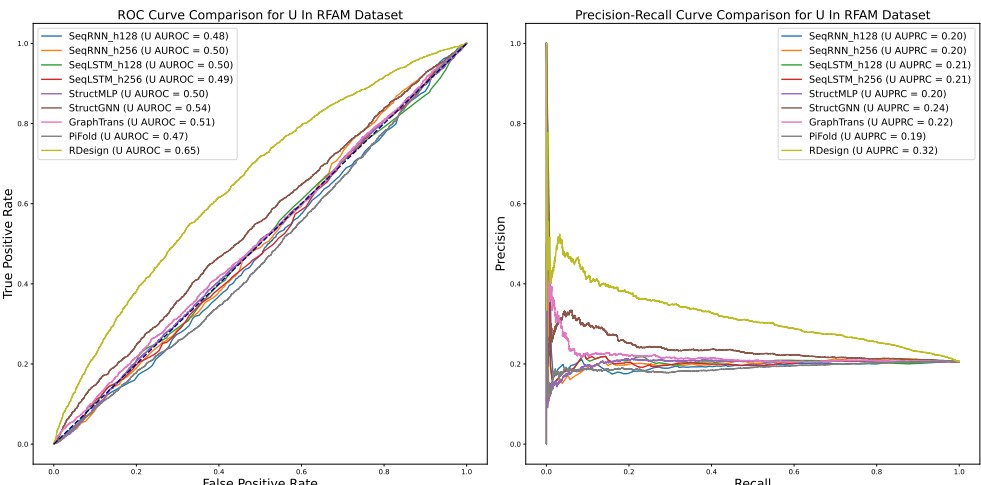

Figure 22: The ROC/PRC curve comparison on the base U of Rfam dataset.

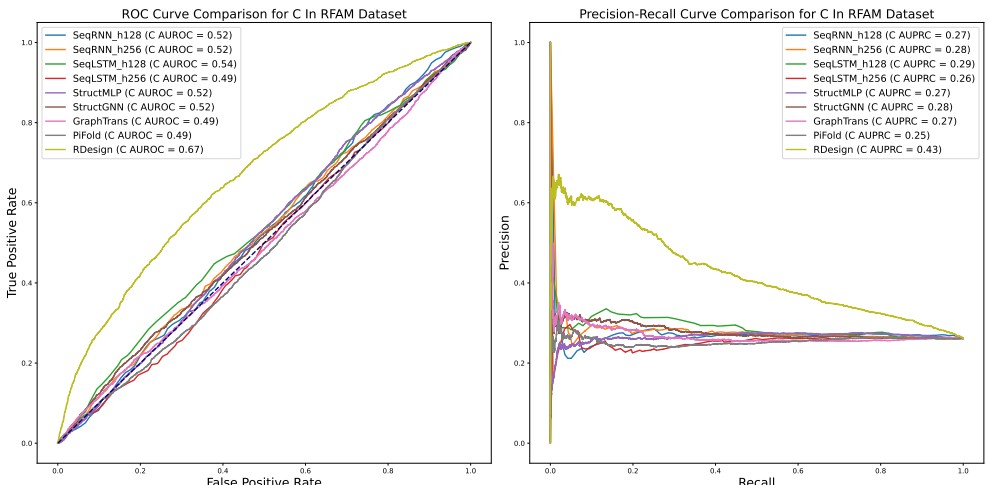

Figure 23: The ROC/PRC curve comparison on the base C of Rfam dataset.

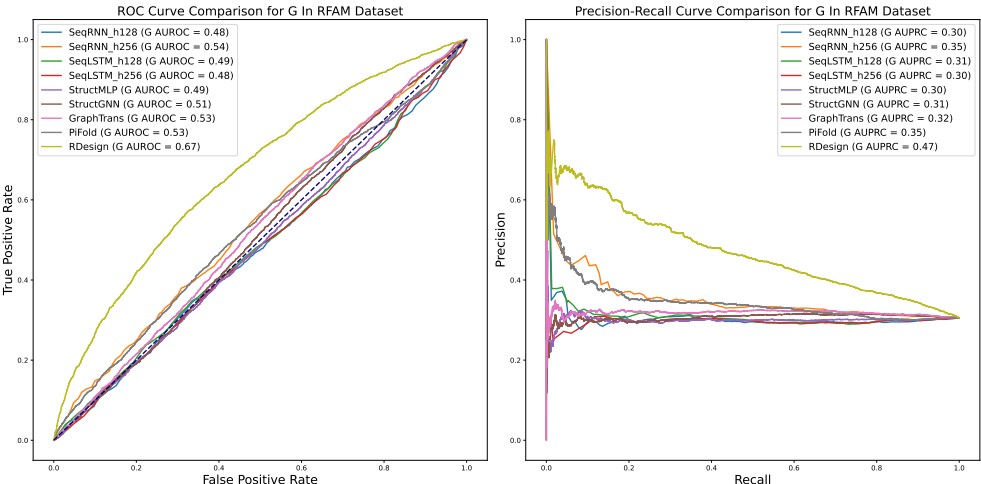

Figure 24: The ROC/PRC curve comparison on the base G of Rfam dataset.

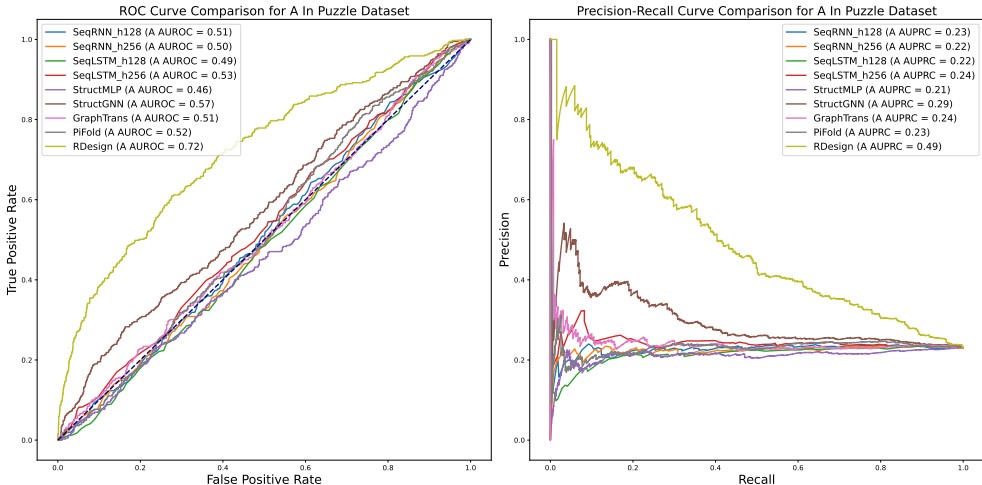

Figure 25: The ROC/PRC curve comparison on the base A of RNA-Puzzles dataset.

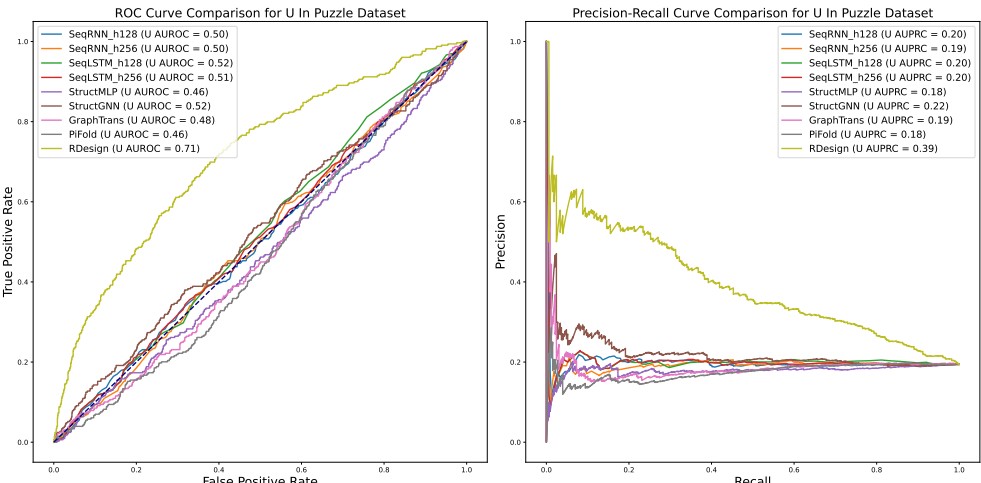

Figure 26: The ROC/PRC curve comparison on the base U of RNA-Puzzles dataset.

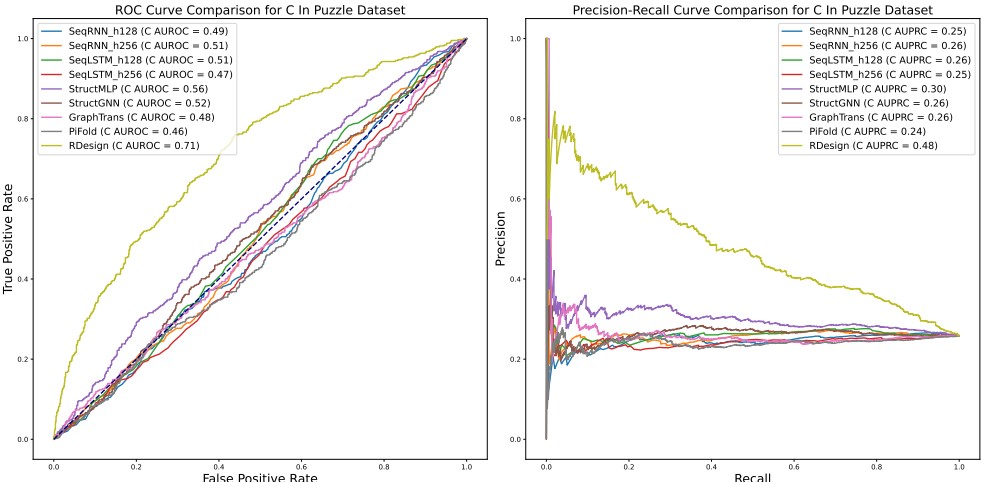

Figure 27: The ROC/PRC curve comparison on the base C of RNA-Puzzles dataset.

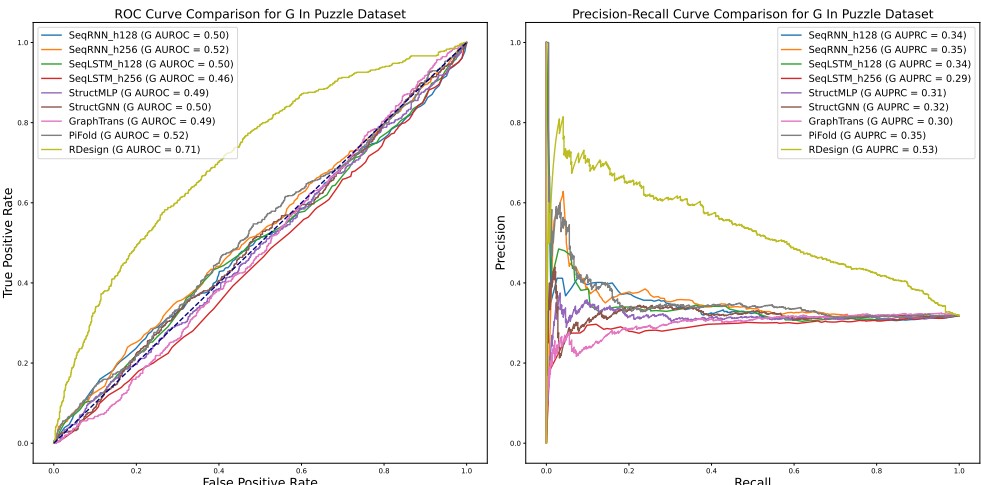

Figure 28: The ROC/PRC curve comparison on the base G of RNA-Puzzles dataset.

