# OpenReview forum: "RDesign: Hierarchical Data-efficient Representation Learning for Tertiary Structure-based RNA Design"
_ICLR.cc/2024/Conference — ICLR 2024 poster_

### Official Review · Reviewer_kDxF · 2023-10-18

**Soundness:** 2 fair
**Presentation:** 1 poor
**Contribution:** 1 poor
**Rating:** 3
**Confidence:** 5

**Summary:**

This paper presents an approach to RNA design that leverages a large, well-curated benchmark dataset and a hierarchical data-efficient representation learning framework.

**Strengths:**

1. The research problem addressed in the article is highly meaningful.
2. The method is easy to follow.

**Weaknesses:**

1. Some parts of the presentation are unclear. For instance, in Section 3.2, it isn't explicitly mentioned until the fourth page that each node represents a nucleotide base and not an atom. Although it becomes clear after reading the entire section, the author should clarify this earlier.
2. The section on "Biomolecular Engineering" in the related work part mainly discusses the significance of RNA design. I believe this content should be integrated into the introduction section rather than being placed under related work. Therefore, the organization of the article could be improved.
3. I find the article lacks novelty; the core multi-level representation learning relies on self-supervised methods from computer vision, which is not novel.
4. The baseline comparisons are incomplete. While the article compares some protein inverse folding methods, it fails to include recent protein inverse folding methods for comparison, such as ProteinMPNN and PiFold, which are mentioned in the related work section.
5. The dataset used in the experiments is not publicly available, making it impossible to directly compare the results with those in previously published works.
6. The author mentions collecting data to create a dataset, which is divided by sequence length. However, this approach is widely used in the protein domain and lacks innovation. Additionally, the author does not cite articles from the protein domain as references here.
7. The author claims to have aggregated and cleaned data from RNAsolo and PDB to create a new benchmark. I find this innovation to be insufficient, and the author does not clearly indicate the limitations of existing datasets or what constitutes "clean data". The motivation behind creating this new benchmark is not convincingly explained.

**Questions:**

My questions correspond to the weaknesses I mentioned above. Additionally, regarding the base pairs mentioned in Section 3.4, apart from the four mentioned, aren't GU and UG pairs also common?

**Minor Concern:**
1. In the "Inter-nucleotide level" section, there is a missing space between the second point and the following content.

---

> ### Author Response · Authors · 2023-11-14
>
> Dear Reviewer kDxF,
>
> We appreciate the effort of reviewers and we always assume that the reviewer is kind and professional. Thus, We sincerely reply to your concerns below.
>
> **Q1** *Benchmark Dataset*
>
> **A1** Our benchmark dataset primarily focuses on two core issues: (i) the scarcity of high-quality data; and (ii) the lack of reasonable split. We have curated a high-quality dataset by manually checking all available data from PDB and RNAsolo. Additionally, we ensured that the distributions of sequence lengths are essentially consistent across the train, validation, and test sets, while simultaneously ensuring that there is no data leakage between them.
>
> **This data partitioning method employs both structural similarity and sequence length distribution, , which we believe differs from the traditional approach of simple partitioning based on sequence length. The specific partitioning process and the analysis of the dataset are detailed in Appendix A and Appendix H, respectively.** If you find similar works that has proposed this partitioning approach, we would be pleased to add a citation.
>
> **Q2** *Dataset Availability*
>
> **A2** We have released the data to the following anonymous link: https://we.tl/t-eU6y2NZLLs. You can download and review it by using the cPickle package in Python.
>
> **Q3** *The baseline comparisons are incomplete. While the article compares some protein inverse folding methods, it fails to include recent protein inverse folding methods for comparison, such as ProteinMPNN and PiFold, which are mentioned in the related work section.*
>
> **A3** Thank you for your suggestion. In this paper, we have selected several classic methods from protein design. From an methodology perspective, ProteinMPNN and StructGNN models are fundamentally similar. However, PiFold introduces a novel approach, utilizing NodeMLP and EdgeMLP, along with Global Context Attention. We have included additional results from PiFold in the revised manuscript. Our findings indicate that while PiFold is effective in protein design, its strong model tends to overfit when applied to data-scarce RNA datasets, resulting in less impressive performance.
>
> **Q4** *Presentation in Manuscript*
>
> **A4** We consider this issue to be quite subjective. Given that Reviewer cP6M evaluated our presentation as "good," and Reviewer J4ia as "excellent," further noting that "overall, the clarity and presentation were significantly above average," we believe that extensive revisions are not necessary.
>
> We agree with your observation that UG and GU are also incorporated in Section 3.4. This was an oversight in our writing. We have fixed this issue in the revised manuscript. In the code implementation, we directly used the dot-bracket format for training, so there are no issues with the experimental results. Thank you very much for your meticulous review!
>
> **Q5** ***I find the article lacks novelty; the core multi-level representation learning relies on self-supervised methods from computer vision, which is not novel.***
>
> **A5** **We would like to know if this is the main reason for your vote for rejection.**
>
> We believe that our primary contributions are: (i) formulating the important biological problem of tertiary structure-based RNA design as one that can be solved by artificial intelligence techniques; (ii) preparing the data for solving this problem by carefully selecting high-quality data and proposing a rigorous partitioning based on structural similarity and sequence length distribution; (iii) addressing the issue of scarce data for this problem by proposing a data-efficient learning framework. **We cannot believe that the use of self-supervised learning methods from computer vision in our data-efficient learning framework is a reasonable reason for rejection.**
>
> By the way, we are curious to know if you would consider increasing the score, or what we would need to do for you to consider raising it.

---

> ### Author Response · Authors · 2023-11-23
> **Waiting for your response**
>
> Dear reviewer kDxF, given the rebuttal deadline is coming soon, I am writing to kindly request your prompt response to our rebuttal. We understand that your time is valuable, but considering the rebuttal efforts made by the authors, we eagerly await your response. We are also reviewers, and from a professional reviewing perspective, your current attitude towards the review is not sufficiently professional:
>
> - Regarding to (1,2,6,7), ICLR allows for revisions, and the textual issues you pointed out are considered minors in other reviews.
>
> - Regarding to (3,6,7), the term "novelty" is often abused, and considering our good performance, it is not sufficient as a reason for rejection. For instance, currently, finetuning large models on CV/NLP can still be accepted as long as the performance is outstanding.
>
> - Regarding to (4,5),we have also included additional experiments on PiFold, and the data links have been provided.
>
> Anyway, we hope you can respond and show your professionalism.

---

### Official Review · Reviewer_J4ia · 2023-10-30

**Soundness:** 3 good
**Presentation:** 4 excellent
**Contribution:** 2 fair
**Rating:** 3
**Confidence:** 4

**Summary:**

This paper presents a method for RNA inverse folding (i.e. given a RNA molecule’s 3D structure, predict the corresponding sequence). Coordinate information about an input molecule’s backbone is fed into a GNN which predicts a nucleotide for each node. Late representations from the GNN are also fed to contrastive losses which contrast structures with slightly perturbed structures, as well as structures with other structures of different folds. The authors demonstrate that this method outperforms other graph-learning methods, and also perform an ablation study to show the usefulness of each loss component (among other attributes).

**Strengths:**

### Interesting and sound architectural design choices

The authors started with a fairly standard GNN architecture, but several modifications and additions on it were made, which reasonably can be expected to improve RNA inverse folding. They adapted the coordinate system from Ingraham, et. al. 2019 to model RNA instead of proteins. The application of contrastive losses at multiple levels (i.e. the cluster-level loss and the sample-level loss) is well-motivated. Additionally, the imposition of secondary-structure constraints via confidence sharpening is a good idea. The ablation study presented also provides good evidence to the usefulness of these additions.

### Overall well-written with many details

Generally, the explanations were clear. Many details were included in the main text or in the supplement which made this paper straightforward to read and understand, which is very appreciated. There were only a few places which I found to be less clear, which will be detailed below. But overall, the clarity and presentation were very above par.

**Weaknesses:**

### Lack of comparison against other RNA inverse-folding methods

I consider this to be the most critical and crucial analysis that is missing. Although the paper does compare the proposed architecture (with its multi-level contrastive losses and secondary-structure loss) to other architectures like LSTMs or graph transformers, there aren’t any comparisons to any other methods which directly tackle the RNA inverse-folding problem. There are _many_ such methods, including several that are as recent as this year (e.g. MCTS-RNA, LEARNA, aRNAque, eM2dRNAs, etc.). Not having these comparisons really holds this paper back, because there is no understanding of how this method (which is being presented as a RNA inverse-folding method) compares to any other RNA inverse-folding method.

### Performance metrics are somewhat limited or measure less important aspects

Although there is some justification provided for the performance metrics focused on in the paper (i.e. recovery and Macro-F1), these metrics are still somewhat limited.

Recovery is simply “accuracy” here, and although this is a great metric to report, some bases will naturally be harder than others to predict. For example, bases participating in certain secondary structures, or those that interact with other residues in the 3D structure, may be easier or harder to classify. Thus, instead of reporting recovery/accuracy pooled over the entire macromolecule, it would be useful to see these numbers stratified by different secondary structures or tertiary attributes.

Macro-F1 is also not particularly clear. It summarizes precision and recall across the 4 different bases and averages them. Since the problem of inverse folding is effectively a classification problem, other more understandable metrics can be used, such as auROC and auPRC (or visualizing the ROC curve and PR curve across different methods). Again, these can be stratified by secondary structure or tertiary attributes.

Additionally, it would be useful to have a more systematic or global analysis of the accuracy of predicted secondary or tertiary structures. Figure 5 is very useful, but those are only a few examples. Having a more global analysis across the entire dataset would be very informative.

### Potential leakage of RNA folds between training and test sets

The process of allocating the training/validation/test sets certainly took into account structural similarity by ensuring that different clusters of structures were allocated to entirely different sets. However, the clusters were generated pretty conservatively (i.e. TM score < 0.45, which is already a very low cut-off). This definition of clusters might be useful for contrastive learning, but it might still let a lot of similar structures to exist in both the training and test set. Instead of reusing the same clusters, it would be good to ensure that the TM scores between training/test sets are much more significantly different (compared to connected components of TM < 0.45), or splitting the training/test sets based on Rfam families to ensure distinct folds. An analysis that shows minimal cross-contamination of folds/structures/scaffolds between the training and test sets would help put this concern to rest.

### Some minor areas to clear up in writing

- The edge features should be $E\in\mathbb{R}^{N\times K\times f_m}$
- Equation 5 is not clear, and does not seem to match typical definitions of MPNNs; for example, it is unclear what the brackets mean and what $h_{E_{ij}}$ is doing on the left; also, it should be $\mathcal{N}(i, K)$ (capital $K$)
- Backwards quote around “reference” in Section 3.4
- In Section 3.4, the confidence scores $c_i$ are not used anywhere; are they equivalent to the model’s output probability? It should also be clarified that $s_i$ in Equation 8 are the true sequence labels

**Questions:**

- Is there a constraint that makes the learned representations more evenly distributed on the hypersphere after projection? The supplement claims that having evenly distributed points on the hypersphere allows the model to better leverage a limited dataset. The projection to hypersphere space just normalizes the magnitude of the representations, but I don’t see how that allows for the distribution of directions on the hypersphere to be uniform.

- Are the dataset splits identical for all baselines?

- Would sharpening confidence be helpful in non-paired residues, as well? It seems like that regularization was added to the paired residues, and the ablation study showed that it was helpful. Would it also be helpful if applied to all residues (not just the paired ones)?

---

> ### Author Response · Authors · 2023-11-14
>
> Dear Reviewer J4ia,
>
> Thanks for your appreciation and detailed review.
>
> **Q1** *Lack of Comparison Against other RNA Invere-folding Methods*
>
> **A1** Thank you for your comment. In this paper, we mainly focus on the tertiary structure-based RNA design. We compare several secondary structure-based RNA design approaches that have public web servers, including RNA Designer, RNAInverse, and INFORNA. The results are shown in the table below. We have added results from secondary structure-based baseline studies to the **Appendix I** of the revised manuscript. If necessary, we will include a discussion about other secondary structure-based approaches.
>
> | Method       | Rfam Recovery (%) | RNA-Puzzles Recovery (%)  | Rfam Macro F1 (×100) | RNA-Puzzles Macro F1 (×100)  |
> |--------------|---------------|--------------------------|--------------------|----------------------------|
> | SeqRNN (h=128) | 27.99±1.21       | 28.99±1.16              | 15.79±1.61           | 16.06±2.02                |
> | SeqRNN (h=256) | 30.94±0.41       | 31.25±0.72              | 13.07±1.57           | 13.24±1.25                |
> | SeqLSTM (h=128)| 24.96±0.46       | 25.78±0.43              | 10.13±1.24           | 10.39±1.50                |
> | SeqLSTM (h=256)| 31.45±0.01       | 31.62±0.20              | 11.76±0.08           | 12.22±0.21                |
> | StructMLP     | 24.40±1.63       | 24.22±1.28              | 16.79±4.01           | 16.40±3.28                |
> | StructGNN     | 27.64±3.31       | 27.96±3.08              | 24.35±3.45           | 22.76±3.19                |
> | GraphTrans    | 23.81±2.57       | 22.21±2.98              | 17.32±5.28           | 17.04±5.36                |
> | **RNAsoft** [1]    | 26.60±5.71       | 28.44±7.31              | 25.86±5.40           | 28.01±7.51                |
> | **RNAInverse** [2]    | 25.46±5.50       | 25.69±6.75              | 24.75±5.50           | 25.01±6.48                |
> | **INFORNA** [3-4]       | 30.45±6.01       | 33.79±9.08              | 27.66±5.62           | 29.94±7.41                |
> | **RDesign**       | **58.25±0.92**   | **50.57±0.81**          | **55.03±0.93**       | **47.06±0.60**            |
>
> **Q2** *Limitation in Performance Mertics*
>
> **A2** Thank you for your insightful comment. Our work primarily focuses on RNA design based on tertiary structure, whereas the performance metrics are largely referenced in protein design. We found that the recovery metric is an intuitive one, while the commonly-used perplexity metric is limited to this work.
>
> Your suggestion is valuable. We have added such evaluation in the **Appendix J**. It can be seen that our approach consistently outperforms the other baselines.
>
> **Q3** *Potential Information Leakage*
>
> **A3** Thank you for pointing out this issue. Data leakage is a longstanding problem in bioinformatics. This is why we strictly split the datasets and provide a detailed dataset analysis in **Appendix H** of the manuscript. We believe our dataset is of high quality and free from any data leakage.
>
> The threshold for TM-Score (0.45) is a common choice in RNA structure prediction research, as proposed by Yang Zhang[5]. In their paper published in Nature Methods, the 0.45 cutoff has been demonstrated to effectively distinguish RNA structures.
>
> **Q4** *Minor areas to clear up*
>
> **A4** Thank you for your careful review. We have refined the manuscript to make it more clear. In Equation 5, the square brackets denote the concatenation operation. The order of the three types of embeddings is not important; here, the edge features between i and j, the node features of i, and the node features of j are concatenated together.
>
> **Q5** *Are the dataset splits identical for all baselines?*
>
> **A5** Yes, we use the same dataset splits for all baselines.
>
> **Q6** *Sharpening Confidence for Non-paired Residues*
>
> **A6** Thank you for your comment. For these non-paired residues, there's no prior knowledge available that could be leveraged to utilize confidence sharpening as a decision-making tool effectively. This distinction in representation between paired and non-paired residues is a key factor in our decision to apply confidence sharpening selectively.
>
> [1] Andronescu, Mirela, et al. "RNAsoft: a suite of RNA secondary structure prediction and design software tools." Nucleic acids research 31.13 (2003): 3416-3422.
>
> [2] Lorenz, Ronny, et al. "ViennaRNA Package 2.0." Algorithms for molecular biology 6 (2011): 1-14.
>
> [3] Disney, Matthew D., et al. "Inforna 2.0: a platform for the sequence-based design of small molecules targeting structured RNAs." ACS chemical biology 11.6 (2016): 1720-1728.
>
> [4] Busch, Anke, and Rolf Backofen. "INFO-RNA—a server for fast inverse RNA folding satisfying sequence constraints." Nucleic acids research 35.suppl_2 (2007): W310-W313.
>
> [5] Chengxin Zhang, Morgan Shine, Anna Marie Pyle, Yang Zhang. US-align: Universal Structure Alignment of Proteins, Nucleic Acids and Macromolecular Complexes. Nature Methods, 19: 1109-1115 (2022)

---

> > ### Comment · Reviewer_J4ia · 2023-11-20
> > **Response to comment**
> >
> > Thank you to the authors for providing these additional results and explanations.
> >
> > **Q1** I sincerely appreciate the additional benchmarks, particularly given the very short time that was available to perform them. Unfortunately, these additional methods are all very outdated. There have been a very large number of RNA inverse-folding methods (many of which also use deep learning) from recent years, which significantly outperform the methods used in Appendix I. In order to show that RDesign is a reasonably competitive method for RNA inverse folding, it is critical to understand how it compares to more recent works.
> >
> > **Q2** I appreciate the stratification of performance by base, but I believe that a much more useful metric is stratification by secondary or tertiary attribute. The main concern here is that different parts of an RNA fold are more difficult to predict. Difficulty may not be correlated to a specific nucleotide, but a more high-level feature like a secondary motif.
> >
> > **Q3** Appendix H has cleared up the concern of potential data leakage for me.

---

> > > ### Author Response · Authors · 2023-11-21
> > >
> > > We are deeply appreciative of the prompt response from the reviewers.
> > >
> > > Our work is not focused on achieving the most advanced secondary structure design, but rather on tertiary structure design. Including additional secondary structure design is beneficial. Given the time constraints, we will endeavor to incorporate additional methods for secondary structure design in our final manuscript.
> > >
> > > In the main text and the appendix, we present comparative analyses across three datasets on five key metrics: recovery, Macro F1, perplexity, AUROC, and AUPRC, thereby offering comprehensive experimental insights. We believe the stratification by structural attribute presents an excellent opportunity for future research.
> > >
> > > Considering the efforts invested during the rebuttal stage, we kindly inquire whether you would consider increasing the score. We would be very grateful for your positive feedback.

---

### Official Review · Reviewer_cP6M · 2023-10-31

**Soundness:** 2 fair
**Presentation:** 3 good
**Contribution:** 3 good
**Rating:** 6
**Confidence:** 4

**Summary:**

This manuscript proposes several methods for identifying the primary structure, or nucleic acid sequence, of an RNA molecule given its tertiary structure, or three-dimensional structure. RNA structure prediction, for reasons that will be discussed, is studied much less than protein structure prediction. This discrepancy presents two major issues: Firstly, there are not enough high-quality RNA structures to build models from, and secondly, there are not as many methods developed to predict RNA structure. Therefore, the authors initially compile and refine a dataset, carefully dividing it into training and testing sets, and then develop new algorithms to address this problem. Since there are no existing methods to predict the RNA sequence from its structure, the authors not only propose a novel method — inspired from Ingraham _et al._ (2019) — but also introduce five baseline methods for comparison. Finally, they test their models on two significant, completely independent datasets: Rfam and RNA-Puzzles.

**Strengths:**

The authors developing this method — the first of its kind — have done an impressive job on multiple fronts:

- Although the proposed method is an extension of the "protein structure to sequence" method proposed in Ingraham et al. (2019), RNA molecules are different from protein molecules, featuring more dihedral angles and a different local geometry. This necessitates the development of new encoding schemes.

- The attention and care the authors have devoted to compiling a clean RNA structure dataset are noteworthy. The training and testing splits are carried out in a structurally-informed manner, which is crucial in structural biology. Additionally, they have maintained similar length distributions.

- Given the limited availability of RNA structures, the authors employ a hierarchical representation learning scheme to address this challenge. This approach groups similar clusters together, much like protein fold families. They also utilize data augmentation by making slight alterations to RNA structures and ensuring their proximity in the latent space.

- The ablation study is informative and serves to justify the modeling decisions made by the authors.

- Impressively, the authors have also developed five baseline models.

**Weaknesses:**

Proteins are arguably not structurally simpler than RNAs, and a reader might get this impression from reading this manuscript. Most RNAs lack a defined structure, and the majority of RNAs do not function primarily through their structures. Even a large number of small non-coding RNAs function through base pairing, such as miRNAs. I believe the authors should demonstrate a tangible application for their proposed method. Is it intended for designing RNA aptamers? If so, how and why would it be superior to SELEX methods?

Minor Issues:
- Figure 4 could be made more clear; the green arrows appear too small. Additionally, clusters and samples within clusters should be better aligned.
- The last sentence of the second paragraph on page 2 is incomplete.

**Questions:**

I decided to move some of the perceived weaknesses to the questions:

- The base atoms are masked; however, purines (R or A/G) and pyrimidines (Y or C/T) differ significantly in size, and the model could have simply learned to predict the base using the spacing between the backbone atoms of neighboring nucleotides. This is not as significant of an issue in proteins, as there are twenty different side chains that do not canonically pair with each other. One way to test this is to evaluate performance by converting all A/G nucleotides to R and all C/T nucleotides to Y, then report recovery scores based on R and Y only. If this score is substantially higher than the original one, then the model may simply be learning whether bases are purines or pyrimidines.

- When constructing the neighborhood graph, wouldn't it be more intuitive to use a ball around each nucleotide? Could enforcing a fixed K result in some edges that are too distant in space?

- Can't this method be iterative? That is, determine the primary sequence from the structure, then compute secondary and tertiary structures from the inferred primary sequence, identify discrepancies, and predict a new primary sequence, repeating this process until a certain condition is met.

- Related to the previous question, is the "recovery score" the best metric? Is the goal to predict THE primary sequence or A primary sequence that gives rise to a very similar tertiary structure? Recovery scores, in general, tend to be low, and reporting predicted RMSD (e.g., using RhoFold) might be helpful in this regard.

- RNA structure is not entirely rigid, and some techniques like NMR can capture an ensemble of structures. Would all such structures in the bundle get the same sequence?

- Approximately 90% of RNA sequences are short (under 100 nt), and a Bayesian Optimization method might work well for this problem. It involves obtaining the tertiary structure, computing the secondary structure, starting with a random sequence, computing its secondary structure, and using the model to propose a new sequence until a close enough solution is found.

- Can Cohen's Kappa be used instead of or in addition to Macro F1?

---

> ### Author Response · Authors · 2023-11-14
>
> Dear Reviewer cP6M,
>
> Thanks for your appreciation and detailed review. We respond to your comments with a heart full of gratitude. We try our best to response the questions below:
>
> **Q1** *Tangible application of RNA design*
>
> **A1** Although for many RNAs, the function is not determined by their tertiary structure, there are some applications for tertiary structure-based RNA design. RNA aptamers are good examples that can bind to specific targets with high affinity and specificity. The SELEX process is a traditional method for identifying aptamers. Our approach may provide several advantages as outlined below:
>
> (i) Data efficiency. Our method proposes a data-efficient way to model RNA structures, particularly beneficial for aptamer design due to the complexity of RNA structures and the vast potential sequence space.
>
> (ii) Structural Precision: By focusing on tertiary structures, our approach can provide a more precise way to design aptamers that specifically fit their targets, potentially outperforming SELEX in precision and fit.
>
> (iii) Speed and Scalability: Computational methods can significantly reduce the time and resources needed for designing effective aptamers compared to iterative lab-based methods such as SELEX.
>
> We believe our approach could be a good alternative to SELEX in scenarios where the traditional method is time-consuming and labor-intensive. Thank you for your valuable and professional suggestions.
>
> **Q2** *Figure and sentence issues*
>
> **A2** Thanks for you kind suggestion. We have updated the figure in the revised manuscript. The sentence you pointed out should be connected with the first sentence of the following paragraph. We have corrected the paragraphing in the revised manuscript.
>
> **Q3** *Atom Spacing Significance*
>
> **A3** In our method, we also consider the information of pairing. In the ablation study, it is observed that the removal of pairing information from the secondary structure results in a slight decrease in performance, but the extent of this decline is relatively limited. Furthermore, StructGNN and GraphTrans, two similar structure-based methods, exhibit poorer performance. Even though such issues might exist, we believe their impact is not significant.
>
> **Q4** *Wouldn't it be more intuitive to use a ball around each nucleotide?*
>
> **A4** In our opinion, using r-ball is similar to k-nearest graph. However, k-nearest is a more convinent way because it always return fix-sized neighbors.
>
> **Q5** *Can't this method be iterative?*
>
> **A5** This method can be easily adopted into an iterative version. In this paper, we identify the key issue for RNA tertiary structure design problem is the sacarce data and we aim to claim that our data-efficient learning manner is beneficial to this problem. The iterative version is supposed to be useful, and we will try to include it in the future work. Thank you for your valuable suggestion!
>
> **Q6** *Reporting predicted RMSD (e.g., using RhoFold)?*
>
> **A6** Thank you for highlighting this aspect. Computing tertiary RNA structures from given sequences is indeed computationally intensive. For instance, assessing over 350 RNA structures across three test sets could potentially require upwards of 1050 hours for completion. Additionally, the current scarcity of available RNA tertiary structures, which serve as labels, introduces an element of bias in RNA structure prediction methods, potentially impacting the accuracy of metrics like RMSD.
>
> Given these constraints and considering our primary objective of accurately predicting RNA sequences, employing the recovery score and macro-f1 score as our main performance metrics are both practical and relevant. These scores directly measures how well our model can regenerate the target RNA sequences, making it a suitable and efficient method to evaluate our model's performance under the current computational and data limitations.

---

> > ### Author Response · Authors · 2023-11-14
> >
> > **Q7** *Would all such structures in the bundle get the same sequence?*
> >
> > **A7** Thank you for your careful review. Our dataset is sourced from RNAcentral, which compiles experimentally determined 3D RNA structures. It cleanses the data of non-RNA elements and categorizes them into equivalent classes. In our dataset, each structure is paired with a unique corresponding sequence. The issue you mentioned does not exist in our dataset.
> >
> > **Q8** *Bayesian Optimization for RNA Sequences*
> >
> > **A8** We always enjoy discussing with professional peers like this reviewer. The idea you presented is impressive. It leverages the method's ability to efficiently search complex spaces to find RNA sequences that correspond to a desired tertiary structure. However, its effectiveness would depend on various factors, including the accuracy of the computational models and the availability of sufficient computational resources. Our proposed approach is likely to be more effective than Bayesian Optimization for capturing the complex geometries and dynamic folding patterns of RNA.
> >
> > **Q9** *Can Cohen's Kappa be used instead of or in addition to Macro F1?*
> >
> > **A9** Thanks for your valuable comment. Cohen's Kappa is commonly used to assess the degree to which two raters or measurement methods agree beyond chance. In the context of RNA design, where the primary goal is to assess the accuracy of nucleotide-level predictions, the recovery score and Macro F1 are more direct measures of performance as they are designed to evaluate the accuracy of multi-class predictions, which is essential in the context of RNA sequence prediction.
> >
> > Indeed, using Cohen's Kappa instead of Macro F1 could potentially offer insights into the consistency of predictions in a different way, focusing on agreement or consistency of the categorization process. However, it might not be as directly relevant to the specific task of RNA design. Therefore, while Cohen's Kappa could provide additional insights, especially in terms of consistency or reliability of predictions, the recovery score and Macro F1 seem more directly aligned with the specific goals and nature of RNA sequence prediction tasks.

---

> > > ### Comment · Reviewer_cP6M · 2023-11-22
> > > **Appreciate the clarifications.**
> > >
> > > I appreciate the authors clarifications and thank them for the time they have spent on drafting their responses. As other reviewers have pointed out, the paper would greatly benefit from more extensive benchmarking.

---

### Author Response · Authors · 2023-11-20

Dear Reviewers,

We sincerely appreciate your time and effort in reviewing our manuscript and offering valuable suggestions.

This year, unlike previous years, there will be no second stage of author-reviewer discussions. A recommendation is required by November 22, 2023. As the author-reviewer discussion phase is drawing to a close, we would like to confirm whether our responses have effectively addressed your concerns.

We provided detailed responses to your concerns a few days ago, and we hope they have adequately addressed your issues. If you require further clarification or have any additional concerns, please do not hesitate to contact us. We are more than willing to continue our communication with you.

Best regards,

The Authors

---

### Meta-Review · Program_Chairs · 2024-01-15

**Metareview:**

The paper presents a method for predicting the RNA sequence from the molecule’s 3D structure. Given the lack of a related benchmark, it introduces a benchmark as well as an algorithm and several baselines. The reviewers acknowledge the validity of the paper’s contributions.
They raised concerns regarding lack of comparisons to recent baselines for secondary structure based RNN inverse design. The authors during the rebuttal showed comparisons against several methods but also highlighted that their paper focuses on the problem of sequence extraction from 3D structure. The authors updated the paper based on reviewers’ comments. The AC thinks the paper does offer a valuable contribution and encourages the authors to add the additional experiments during rebuttal time to the camera ready version.

**Justification For Why Not Higher Score:**

The paper’s empirical analysis could be improved, as indicated by the reviewers’ comments.

**Justification For Why Not Lower Score:**

The paper offers both a new dataset and an algorithm in an under researched problem.

---

### Decision · Program_Chairs · 2024-01-16

Accept (poster)